# Inositol pyrophosphate dynamics reveals control of the yeast phosphate starvation program through 1,5-IP$_8$ and the SPX domain of Pho81

**Valentin Chabert[1†], Geun-Don Kim[1†], Danye Qiu[2], Guizhen Liu[2], Lydie Michaillat Mayer[1], Muhammed Jamsheer K[1], Henning J Jessen[2], Andreas Mayer[1]***

[1]Département d'immunobiologie, Université de Lausanne, Epalinges, Switzerland; [2]Institute of Organic Chemistry, Centre for Integrative Biological Signalling Studies, University of Freiburg, Freiburg, Germany

**\*For correspondence:**
andreas.mayer@unil.ch

[†]These authors contributed equally to this work

**Competing interest:** The authors declare that no competing interests exist.

**Abstract** Eukaryotic cells control inorganic phosphate to balance its role as essential macronutrient with its negative bioenergetic impact on reactions liberating phosphate. Phosphate homeostasis depends on the conserved INPHORS signaling pathway that utilizes inositol pyrophosphates and SPX receptor domains. Since cells synthesize various inositol pyrophosphates and SPX domains bind them promiscuously, it is unclear whether a specific inositol pyrophosphate regulates SPX domains in vivo, or whether multiple inositol pyrophosphates act as a pool. In contrast to previous models, which postulated that phosphate starvation is signaled by increased production of the inositol pyrophosphate 1-IP$_7$, we now show that the levels of all detectable inositol pyrophosphates of yeast, 1-IP$_7$, 5-IP$_7$, and 1,5-IP$_8$, strongly decline upon phosphate starvation. Among these, specifically the decline of 1,5-IP$_8$ triggers the transcriptional phosphate starvation response, the PHO pathway. 1,5-IP$_8$ inactivates the cyclin-dependent kinase inhibitor Pho81 through its SPX domain. This stimulates the cyclin-dependent kinase Pho85-Pho80 to phosphorylate the transcription factor Pho4 and repress the PHO pathway. Combining our results with observations from other systems, we propose a unified model where 1,5-IP$_8$ signals cytosolic phosphate abundance to SPX proteins in fungi, plants, and mammals. Its absence triggers starvation responses.

## eLife assessment

This **fundamental** study describes the mechanisms for regulation of the phosphate starvation response in baker's yeast, clarifies the interpretations of prior data, and suggests a unifying mechanism across eukaryotes. The study provides **compelling** data, based on biochemical analyses, protein localization by fluorescence, and genetic approaches that 1,5-InsP8 is the phosphate nutrient messenger in yeast.

## Introduction

Inorganic phosphate (P$_i$) is an essential nutrient for all living systems. While required in large amounts for synthesis of nucleic acids, phospholipids, and phosphorylated carbohydrates and proteins, an

overaccumulation of $P_i$ decreases the free energy provided by $P_i$-liberating reactions, such as nucleotide hydrolysis, which might stall metabolism (*Austin and Mayer, 2020*). In fungi, plants, and animals, control of $P_i$ homeostasis involves *myo*-inositol pyrophosphates and a family of evolutionarily conserved SPX domains, constituting the core of a postulated signaling pathway that we termed INPHORS (*Austin and Mayer, 2020*; *Azevedo and Saiardi, 2017*).

SPX domains interact with or form part of a large variety of proteins that affect $P_i$ homeostasis by transporting $P_i$ across membranes, converting it into polyphosphates (polyP) or other metabolites, or regulating $P_i$-dependent transcription (*Secco et al., 2012*). Direct regulation of an SPX-containing protein by synthetic inositol pyrophosphates was shown for the polyP polymerase VTC, the $P_i$ transporters Pho91 and the PHR transcription factors in plants (*Gerasimaite et al., 2017*; *Potapenko et al., 2018*; *Wild et al., 2016*; *Ried et al., 2021*; *Wang et al., 2015*; *Liu et al., 2016*; *Dong et al., 2019*; *Liu et al., 2023*; *Pipercevic et al., 2023*; *Guan et al., 2023*). A firm link suggesting that the SPX domain as the receptor for inositol pyrophosphate regulation was provided by point mutants in the SPX domain that rendered VTC either independent of activation by inositol pyrophosphates or non-responsive to them (*Wild et al., 2016*). Structural analysis revealed that many of these mutations localized to a highly charged region that can bind inositol poly- and pyrophosphates with high affinity. However, this binding site discriminates poorly between different inositol pyrophosphates at the level of binding (*Wild et al., 2016*). By contrast, strong differences are observed in the agonist properties, leading to the suggestion that binding affinity is a poor predictor of inositol pyrophosphate specificity and activity (*Austin and Mayer, 2020*; *Gerasimaite et al., 2017*; *Pipercevic et al., 2023*).

Inositol pyrophosphates that can be found in a wide variety of organisms carry seven ($IP_7$) or eight ($IP_8$) phosphates occupying every position around myo-inositol ring. The $IP_7$ isomers $1PP\text{-}InsP_5$ and $5PP\text{-}InsP_5$ carry diphosphate groups at the 1- or 5-position, respectively, and the $1,5(PP)_2\text{-}InsP_4$ carries two diphosphate groups at the 1- and 5-position. For convenience, we refer to these inositol pyrophosphates as $5\text{-}IP_7$, $1\text{-}IP_7$, and $1,5\text{-}IP_8$ from hereon. All three inositol pyrophosphates are linked to phosphate homeostasis by the fact that genetic ablation of the enzymes making them, such as IP6Ks (inositol hexakisphosphate kinases), PPIP5Ks (diphosphoinositol pentakisphosphate kinases), and ITPKs (inositol tris/tetrakisphosphate kinases), alters the phosphate starvation response. Considerable discrepancies exist in the assignment of these inositol pyrophosphates to different aspects of $P_i$ homeostasis. In mammalian cells, $1,5\text{-}IP_8$ activates the only SPX protein expressed in this system, the $P_i$ exporter XPR1 (*Li et al., 2020*; *Wilson et al., 2019*). $1,5\text{-}IP_8$ was also proposed as a regulator of the phosphate starvation response in *Arabidopsis* because $1,5\text{-}IP_8$, but not $5\text{-}IP_7$, promotes the interaction of SPX1 with the $P_i$-responsive transcription factor PHR1 in vitro (*Dong et al., 2019*). On the other hand, quantitative measurements of the interaction of inositol pyrophosphates with rice SPX4 and its cognate $P_i$-responsive transcription factor PHR2 revealed only a minor, twofold difference in the $K_d$ values for $5\text{-}IP_7$ and $1,5\text{-}IP_8$. $1\text{-}IP_7$, $5\text{-}IP_7$, and $1,5\text{-}IP_8$ all decrease upon $P_i$ starvation and mutants lacking either of the enzymes necessary for their synthesis in plants, the ITPKs and the PPIP5Ks, induce the phosphate starvation response (*Dong et al., 2019*; *Riemer et al., 2021*; *Laha et al., 2015*; *Zhu et al., 2019*). Analysis of respective *Arabidopsis* mutants revealed that $P_i$ concentration in shoots correlates poorly with their content of $IP_7$ and $IP_8$ (*Riemer et al., 2021*). While *itpk1* mutants, lacking one of the enzymes that generates $5\text{-}IP_7$, have similar $1,5\text{-}IP_8$ content as wildtype, their $P_i$ content is almost twofold increased and their $5\text{-}IP_7$ content is reduced by a factor of four. Inversely, mutants lacking the PPIP5K VIH2 show a fivefold reduction of $1,5\text{-}IP_8$ but normal $P_i$ content (*Riemer et al., 2021*). Although $1,5\text{-}IP_8$ is the inositol pyrophosphate that is the most responsive to $P_i$ starvation and $P_i$ re-feeding (*Riemer et al., 2021*) this has rendered it difficult to clearly resolve whether $IP_7$, $IP_8$, or both signal cellular $P_i$ status to the $P_i$ starvation program in vivo. That plants are composed of multiple source and sink tissues for phosphate may complicate the analysis, because systemic knockouts of inositol pyrophosphate synthesizing enzymes may exert their main effect in a tissue that is different from the one where the starvation response is scored.

Pioneering studies on the regulation of the $P_i$ starvation program of *Saccharomyces cerevisiae*, the PHO pathway, concluded that this transcriptional response is triggered by an increase in $1\text{-}IP_7$ (*Lee et al., 2007*; *Lee et al., 2008*), whereas subsequent studies in the pathogenic yeast *Cryptococcus neoformans* proposed $5\text{-}IP_7$ as the necessary signaling compound (*Desmarini et al., 2020*). Studies using the *S. cerevisiae* polyphosphate polymerase VTC as a model suggested that this enzyme, which is necessary for polyP accumulation under $P_i$-replete conditions, is stimulated by $5\text{-}IP_7$ in vivo

(*Gerasimaite et al., 2017*; *Liu et al., 2023*; *Lonetti et al., 2011*; *Auesukaree et al., 2005*), whereas studies in *Schizosaccharomyces pombe* proposed IP$_8$ as the stimulator (*Pascual-Ortiz et al., 2021*). *S. pombe* has similar enzymes for IP$_8$ synthesis and hydrolysis as *S. cerevisiae* (*Zhai et al., 2015*; *Randall et al., 2020*; *Pascual-Ortiz et al., 2018*; *Topolski et al., 2016*; *Pöhlmann et al., 2014*). In contrast to *S. cerevisiae*, genetic ablation of 1-IP$_7$ and IP$_8$ production in *S. pombe* leads to hyper-repression of the transcriptional phosphate starvation response (*Pascual-Ortiz et al., 2018*; *Sanchez et al., 2019*; *Benjamin et al., 2022*) and interferes with the induction of the transcriptional phosphate starvation response. However, the transcriptional phosphate starvation response in *S. pombe* occurs through a different set of protein mediators. For example, it lacks homologs of Pho81 and uses Csk1 instead of Pho85-Pho80 for phosphate-dependent transcriptional regulation (*Estill et al., 2015*; *Henry et al., 2011*; *Carter-O'Connell et al., 2012*; *Schwer et al., 2021*). Furthermore, PHO pathway promotors in *S. pombe* overlap with and are strongly regulated by lncRNA transcription units (*Sanchez et al., 2018*; *Schwer et al., 2015*; *Schwer et al., 2014*). It is hence difficult to compare the downstream events in this system to the PHO pathway of *S. cerevisiae*.

The discrepancies mentioned above could either reflect a true divergence in the signaling properties of different inositol pyrophosphates in different organisms, in which case a common and evolutionarily conserved signaling mechanism may not exist. Alternatively, the diverging interpretations could reflect limitations in the analytics of inositol pyrophosphates and in the in vivo assays for the fundamental processes of P$_i$ homeostasis. The analysis of inositol pyrophosphates is indeed very challenging in numerous ways. Their cellular concentrations are very low, the molecules are highly charged, and they exist in multiple isomers that differ only in the positioning of the pyrophosphate groups. Inositol pyrophosphate analysis has traditionally been performed by ion exchange HPLC of extracts from cells radiolabeled through $^3$H-inositol (*Wilson and Saiardi, 2017*). This approach requires constraining and slow labeling schemes, is costly and time-consuming. Furthermore, HPLC-based approaches in most cases did not resolve isomers of IP$_7$ or IP$_8$. These factors severely limited the number of samples and conditions that could be processed and the resolution of the experiments.

The recent use of capillary electrophoresis coupled to mass spectrometry (CE-MS) has dramatically improved the situation, permitting resolution of many regio-isomers of IP$_7$ and IP$_8$ without radiolabeling, and at superior sensitivity and throughput. We harnessed the potential of this method to dissect the role of the three known inositol pyrophosphates that accumulate in *S. cerevisiae*. We analyzed their impact on the PHO pathway, which is a paradigm for a P$_i$-controlled transcriptional response and the regulation of phosphate homeostasis (*Austin and Mayer, 2020*; *Eskes et al., 2018*; *Conrad et al., 2014*). Beyond this physiological function, however, the PHO pathway also gained widespread recognition as a model for promotor activation, transcription initiation, chromatin remodeling, and nucleosome positioning (*Korber and Barbaric, 2014*). In the PHO pathway, the cyclin-dependent kinase inhibitor (CKI) Pho81 translates intracellular P$_i$ availability into an activation of the cyclin-dependent kinase (CDK) complex Pho85-Pho80 (*Lee et al., 2007*; *Schneider et al., 1994*; *Ogawa et al., 1995*; *Yoshida et al., 1989*). At high P$_i$, Pho85-Pho80 phosphorylates and inactivates the key transcription factor of the PHO pathway, Pho4, which then accumulates in the cytosol (*Kaffman et al., 1998b*; *Kaffman et al., 1998a*; *Komeili and O'Shea, 1999*). During P$_i$ starvation, Pho81 inhibits the Pho85-Pho80 kinase, leading to dephosphorylation and activation of Pho4 and the ensuing expression of P$_i$-responsive genes (PHO genes).

The activation of Pho85-Pho80 through Pho81 has been explored in detail, leading to a series of highly influential studies that have gained wide acceptance in the field. A critical function was ascribed to 1-IP$_7$ in activating the CDK inhibitor PHO81, allowing it to inhibit Pho85-Pho80 (*Lee et al., 2007*). IP$_7$ concentration was reported to increase upon P$_i$ starvation, and this increase was considered as necessary and sufficient to inactivate Pho85-Pho80 kinase through Pho81 and trigger the PHO pathway. The 1-IP$_7$ binding site on Pho81 was mapped to a short stretch of 80 amino acids, the 'minimum domain' (*Lee et al., 2007*; *Lee et al., 2008*; *Ogawa et al., 1995*; *Huang et al., 2001*). This domain is in the central region and distinct from the N-terminal SPX domain of Pho81. Since overexpression of the minimum domain rescued P$_i$-dependent regulation of the PHO pathway to some degree, the key regulatory function was ascribed to this minimum domain and the SPX domain was considered as of minor importance. By contrast, earlier studies based on unbiased mutagenesis, which subsequently received much less attention, had identified mutations in other regions of PHO81 with significant impact on PHO pathway activation (*Ogawa et al., 1995*; *Spain et al., 1995*; *Toh-E*

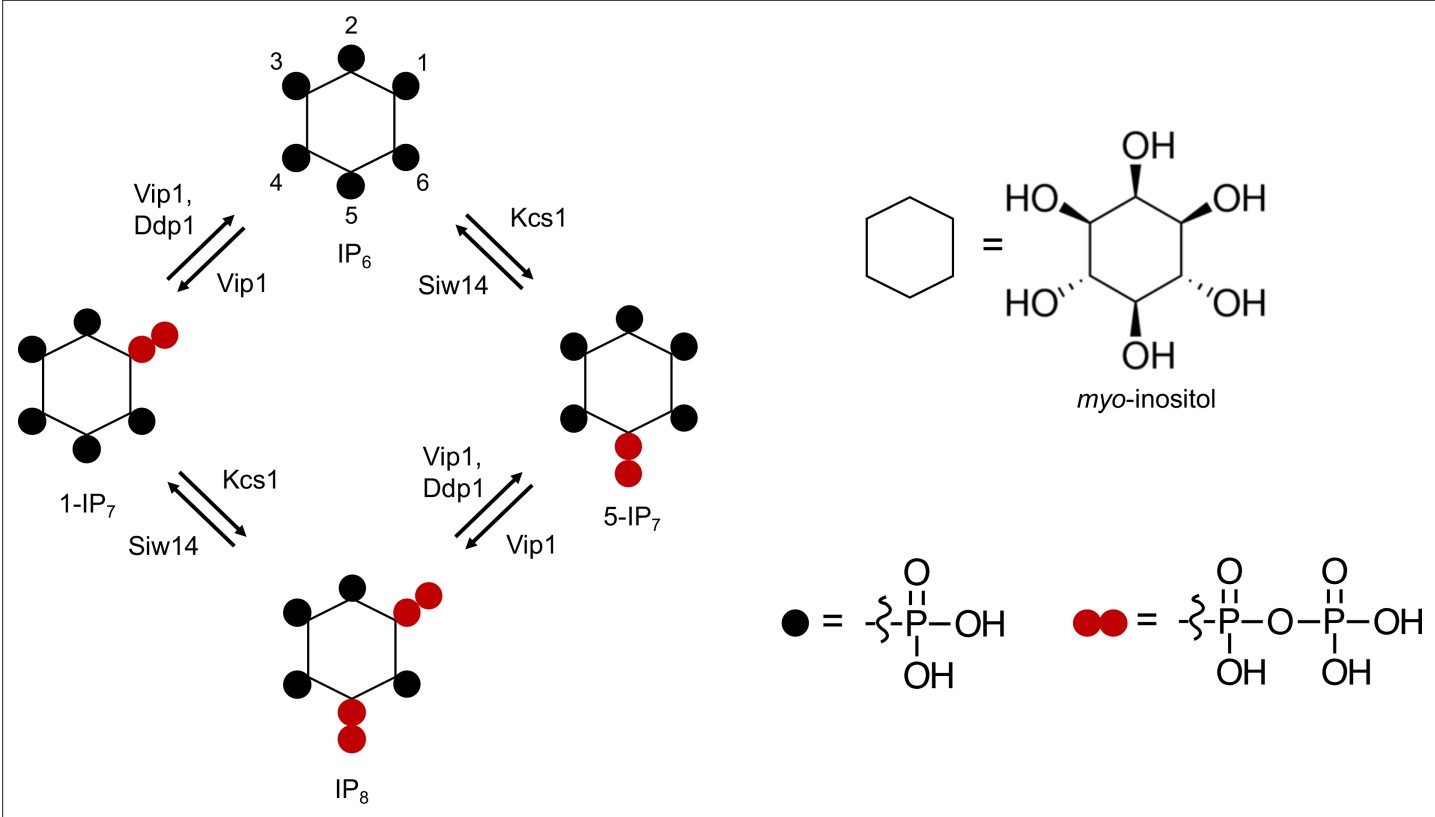

**Figure 1.** Pathways of inositol pyrophosphate metabolism in *S. cerevisiae*.

*and Oshima, 1974*; *Creasy et al., 1993*; *Creasy et al., 1996*). The situation is complicated by further studies, which found global $IP_7$ levels to decrease rather than increase upon $P_i$ starvation (*Wild et al., 2016*; *Li et al., 2020*; *Lonetti et al., 2011*; *Kim et al., 2023*). The analytics used in most studies could not distinguish $1\text{-}IP_7$ and $5\text{-}IP_7$, however, leaving open the possibility that an increase of $1\text{-}IP_7$ might be masked by a decrease of a much larger pool of $5\text{-}IP_7$.

To resolve these discrepancies and revisit the regulation of the PHO pathway by inositol pyrophosphates and Pho81, we capitalized on recent advances in the non-radioactive analysis of inositol pyrophosphates through CE-MS, offering superior resolution, sensitivity, and throughput (*Qiu et al., 2020*; *Qiu et al., 2023*). We combined comprehensive analyses of PHO pathway activation and inositol pyrophosphate profiles in mutants of key enzymes involved in inositol pyrophosphate metabolism of yeast to determine the inositol pyrophosphate species relevant to PHO pathway control and their impact on Pho85-Pho80 kinase. This led us to a revised model of PHO pathway regulation.

## Results

So far, analysis of the role of inositol pyrophosphates in $P_i$ homeostasis and $P_i$ starvation responses relied mainly on the use of mutants which ablate one of the pathways of inositol pyrophosphate synthesis (*Figure 1*). Yet, as shown in plants, several inositol pyrophosphates can change in a similar manner upon $P_i$ depletion or replenishment, and ablation of enzymes adding phosphate groups at the 1- or 5-positions of the inositol ring induces similar $P_i$ starvation responses (*Dong et al., 2019*; *Riemer et al., 2021*). This renders it difficult to distinguish a role for an individual inositol pyrophosphate from the alternative hypothesis that all inositol pyrophosphates collectively contribute to signaling. To dissect this issue in yeast, we performed time course analyses of mutants in inositol pyrophosphate phosphatases and kinases, in which we correlate the levels of all inositol pyrophosphates with the induction of the phosphate starvation response in yeast. The rationale was to seek for upper or lower thresholds of inositol pyrophosphate concentrations during PHO pathway induction and using those to sieve out the inositol pyrophosphate responsible for signaling $P_i$ starvation.

## Dynamics of the cytosolic concentrations of 5-IP$_7$, 1-IP$_7$, and 1,5-IP$_8$

In yeast, the *myo*-inositol hexakisphosphate kinases Vip1 and Kcs1 generate 1-IP$_7$ and 5-IP$_7$, respectively, and they are both required for synthesis of 1,5-IP$_8$ (*Figure 1*; *Mulugu et al., 2007*; *Saiardi et al., 1999*; *Zong et al., 2022*; *Wang et al., 2012*; *Zong et al., 2021*). Inositol pyrophosphatases, such as Ddp1 and Siw14, dephosphorylate these compounds at the 1- and 5-position, respectively (*Figure 1A*; *Lonetti et al., 2011*; *Wang et al., 2018*; *Steidle et al., 2020*). We analyzed the kinetics and the role of these inositol pyrophosphates in PHO pathway activation. To this end, yeasts were cultured in liquid synthetic complete (SC) media to early logarithmic phase and then transferred to P$_i$-free SC medium. The cells were extracted with perchloric acid and inositol phosphates were analyzed by capillary electrophoresis coupled to electrospray ionization (ESI) mass spectrometry (*Qiu et al., 2020*). Three inositol pyrophosphates were detectable: 1-IP$_7$, 5-IP$_7$, and 1,5-IP$_8$. We note that the CE-MS approach does not differentiate pyrophosphorylation of the inositol ring at the 1- and 3-positions. Thus, our assignments of the relevant species as 1-IP$_7$ and 1,5-IP$_8$ are based on previous characterization of the reaction products and specificities of IP6Ks and PPIPKs (*Mulugu et al., 2007*; *Saiardi et al., 1999*; *Zong et al., 2022*; *Wang et al., 2012*; *Zong et al., 2021*; *Dollins et al., 2020*).

Quantitation of inositol pyrophosphates by CE-MS was facilitated by spiking the samples with synthetic, [13]C-labeled inositol pyrophosphate standards (*Harmel, 2019*; *Puschmann et al., 2019*). The recovery rate of inositol pyrophosphates during the extraction was determined by adding known quantities of synthetic standards to the cells already before the extractions. This demonstrated that 89% of 1-IP$_7$, 90% of 5-IP$_7$, and 75% of 1,5-IP$_8$ were recovered in the extract (*Figure 2—figure supplement 1*). To estimate the cellular concentrations of these compounds, the volume of the cells was determined by fluorescence microscopy after staining of the cell wall with trypan blue (*Figure 2—figure supplement 2*). This yielded an average cell volume of 42 fL. Detailed morphometric studies of yeast showed that the nucleus occupies around 8% of this volume and that all other organelles collectively account for approx. 18% (*Uchida et al., 2011*). Taking this into account we can estimate the concentrations in the cytosolic space (including the nucleus, which is permeable to small molecules) of cells growing logarithmically on SC medium as 0.5 µM for 1-IP$_7$, 0.7 µM for 5-IP$_7$, and 0.3 µM for 1,5-IP$_8$ (*Figure 2A*).

Next, we determined the impact of Kcs1, Vip1, Siw14, and Ddp1 on the inositol pyrophosphate levels in the cells (*Figure 2A*). 5-IP$_7$ was not detected in the *kcs1Δ* mutant and 1-IP$_7$ was strongly reduced in the *vip1Δ* strain. 1,5-IP$_8$ was undetectable in *kcs1Δ* and decreased by 75% in *vip1Δ*. The nature of the residual 1,5-IP$_8$ and 1-IP$_7$ signals is currently unclear. They may represent inositol pyrophosphates synthesized by enzymes other than Kcs1 and Vip1, such as the inositol polyphosphate multi-kinases, which can also produce inositol pyrophosphates (*Riemer et al., 2021*; *Zong et al., 2022*; *Laha et al., 2019*; *Adepoju et al., 2019*; *Whitfield et al., 2020*). Importantly, residual 1,5-IP$_8$ and 1-IP$_7$ were not observed in P$_i$-starved wildtype cells (*Figure 2B*). This may be due to presence of the Vip1 phosphatase activity, which is missing in *vip1Δ* cells, but which may quench weak production of inositol pyrophosphates such as 1-IP$_7$ or 1,5-IP$_8$ by other enzymes in wildtype cells. Since this aspect is not central to the question of our study, it was not pursued further. *kcs1Δ* mutants showed a two- to threefold decrease in 1-IP$_7$, suggesting that the accumulation of 1-IP$_7$ depends on 5-IP$_7$. This might be explained by assuming that, in the wildtype, most 1-IP$_7$ stems from the conversion of 5-IP$_7$ to 1,5-IP$_8$, followed by dephosphorylation of 1,5-IP$_8$ to 1-IP$_7$. A systematic analysis of this interdependency will require rapid pulse-labeling approaches for following the turnover of the phosphate groups, which are not yet established for inositol pyrophosphates (*Wilson and Saiardi, 2017*; *Harmel, 2019*; *Nguyen Trung et al., 2022*; *Azevedo and Saiardi, 2006*). An unexpected finding was the up to 20-fold overaccumulation of 5-IP$_7$ in the *vip1Δ* mutant. By contrast, *ddp1Δ* cells showed normal levels of 5-IP$_7$ and 1,5-IP$_8$ but a 10-fold increase in 1-IP$_7$. *siw14Δ* cells showed a fivefold increase in 5-IP$_7$, but similar levels of 1,5-IP$_8$ and 1-IP$_7$ as wildtype.

We performed time course experiments to analyze how inositol pyrophosphate levels change under P$_i$ withdrawal. 5-IP$_7$ was the predominant inositol pyrophosphate species in wildtype cells growing on P$_i$-replete media (*Figure 2B*). Within 30 min of P$_i$ starvation, the concentration of all three inositol pyrophosphate species rapidly decreased by 75% for 1,5-IP$_8$, by 47% for 1-IP$_7$, and by 40% for 5-IP$_7$. This decline continued, so that 1-IP$_7$ and 1,5-IP$_8$ became undetectable and only 3% of 5-IP$_7$ remained after 4 hr, corresponding to a concentration below 50 nM. The high excess of 5-IP$_7$ in *vip1Δ* cells also declined as soon as the cells were transferred to P$_i$ starvation medium (*Figure 2C*).

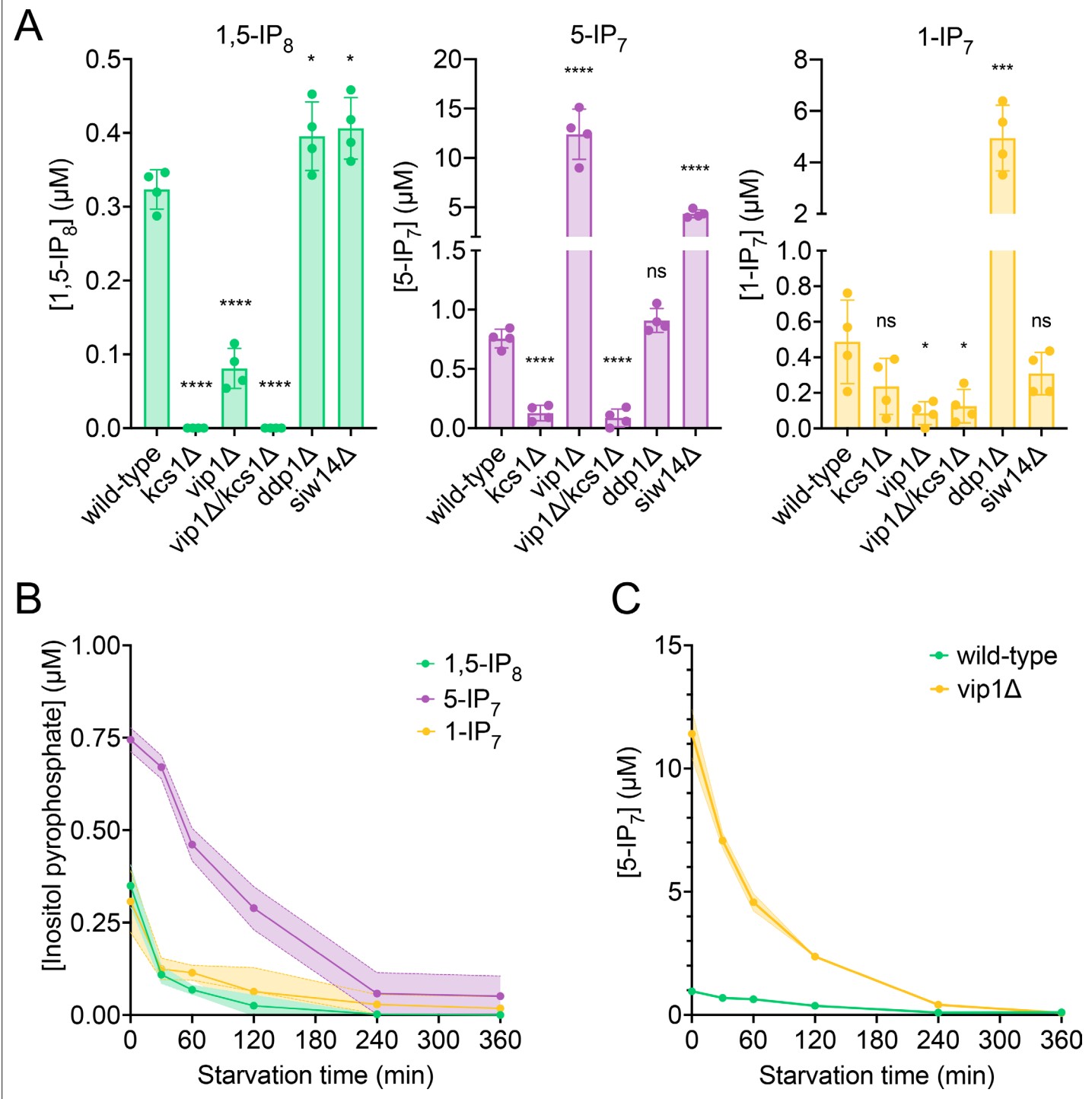

**Figure 2.** Cytosolic concentrations of 5-IP$_7$, 1-IP$_7$, and 1,5-IP$_8$. (**A**) Inositol pyrophosphate concentrations in the cytosol. The indicated strains were grown logarithmically in synthetic complete (SC) medium containing 7.5 mM of inorganic phosphate (P$_i$) (30°C, 150 rpm, overnight). When OD$_{600nm}$ reached 1 (1 × 10$^7$ cells/mL), 1 mL of culture was extracted with perchloric acid and analyzed for inositol pyrophosphates by CE-ESI-MS. The y-axis provides the estimated cytosolic concentrations based on an average cell volume of 42 fL. Means (n=4) and standard deviations are indicated. ****p<0.0001; ***p<0.001; **p<0.01; *p<0.05; n.s. not significant, tested with Student's t-test. (**B**) Evolution of inositol pyrophosphate species during P$_i$ starvation. Cells were grown as in A, washed twice with P$_i$ starvation medium, and further incubated in P$_i$ starvation medium. The inoculum for the samples bound to be extracted after different times of further incubation in starvation medium was adjusted such that all samples had similar OD$_{600nm}$ at the time of harvesting (OD$_{600nm}$=0.5 for 30 min and 60 min samples; OD$_{600nm}$=0.4 for 120 min and 240 min samples; OD$_{600nm}$=0.25 for 360 min samples). At the indicated times in starvation media, 1 mL aliquots were extracted and analyzed for inositol pyrophosphates as in A. The data was normalized by the number of cells

*Figure 2 continued on next page*

*Figure 2 continued*

harvested before calculating cytosolic concentrations. Means and standard deviations are given (n=3). (**C**) Depletion of 5-IP$_7$ in starving *vip1Δ* cells. The indicated cells were grown in Pi-replete medium and then transferred to Pi starvation medium as in B. At the indicated times, samples were extracted and analyzed for 5-IP$_7$ as in A. Means and standard deviations (n=4) are shown as solid lines and shaded areas, respectively.

The online version of this article includes the following figure supplement(s) for figure 2:

**Figure supplement 1.** Recovery of inositol pyrophosphates extracted from *S. cerevisiae* cells.

**Figure supplement 2.** Determination of cell dimensions from wildtype cells stained with trypan blue.

**Figure supplement 3.** Loss of inositol pyrophosphates from *siw14Δ* cells upon inorganic phosphate (P$_i$) starvation.

After 2 hr of starvation, it was still twofold above the concentration measured in P$_i$-replete wildtype cells. Even after 3.5–4 hr, P$_i$-starved *vip1Δ* cells had just reached the 5-IP$_7$ concentration of P$_i$-replete wildtype cells. A comparable decline of all inositol pyrophosphate species upon P$_i$ starvation could be observed in other fungi, such as *S. pombe* and *C. neoformans* (**Figure 3**), suggesting that this response is conserved.

Taken together, P$_i$ starvation leads to a virtually complete depletion of all three inositol pyrophosphate species, with 1,5-IP$_8$ declining faster than 1-IP$_7$ and 5-IP$_7$. Furthermore, inositol pyrophosphatase mutants provide the possibility to generate relatively selective increases in 5-IP$_7$ and 1-IP$_7$. We used this information to dissect the impact of 5-IP$_7$, 1-IP$_7$, and 1,5-IP$_8$ on control of the PHO pathway.

## 1,5-IP$_8$ signals cytosolic P$_i$ levels to the PHO pathway

To this end, we correlated the measured inositol pyrophosphate concentrations to the induction of the PHO pathway. We assayed a key event of PHO pathway activation, partitioning of the fluorescently tagged transcription factor Pho4$^{yEGFP}$ between the cytosol and the nucleus. Pho4 shuttles between nucleus and cytosol and its phosphorylation through Pho85-Pho80 favors Pho4 accumulation in the cytosol. The relocation of Pho4 can hence serve as an in vivo indicator of PHO pathway activation (**Thomas and O'Shea, 2005**; **Wykoff et al., 2007**; **O'Neill et al., 1996**). It provides a readout for Pho85-Pho80 activity. *PHO4* was tagged at its genomic locus, making Pho4$^{yEGFP}$ the sole source of this transcription factor. Pho4 relocation was assayed through automated image segmentation and analysis. The artificial intelligence-based segmentation algorithm recognized more than 90% of the cells in a bright-field image and delimited their nuclei based on a red-fluorescent nuclear mCherry marker (**Figure 4—figure supplement 1**). This segmentation allows quantitative measurements of Pho4 distribution between the cytosol and nucleus in large numbers of cells. In addition, we assayed PHO pathway activation through fluorescent yEGFP reporters expressed from the *PHO5* (*prPHO5-yEGFP*) and *PHO84* (*prPHO84-yEGFP*) promotors. These are classical assays of PHO pathway activation, but their output is further downstream and hence comprises additional regulation, for example at the level of chromatin or RNA, or the direct activation of Pho4 through metabolites such as AICAR (**Nishizawa et al., 2008**; **Almer et al., 1986**; **Barbaric et al., 2007**; **Lam et al., 2008**; **Pinson et al., 2009**). Upon P$_i$ withdrawal, both promotors are induced by the PHO pathway but the *PHO84* promotor reacts in a more sensitive manner and is induced more rapidly than the *PHO5* promotor (**Thomas and O'Shea, 2005**).

In wildtype cells grown under P$_i$-replete conditions, Pho4$^{yEGFP}$ was cytosolic and the *PHO5* and *PHO84* promotors were inactive, indicating that the PHO pathway was repressed (**Figure 4**). Within 30 min of starvation in P$_i$-free medium, Pho4$^{yEGFP}$ relocated into the nucleus and *PHO84* and *PHO5* were strongly induced. By contrast, *kcs1Δ* cells showed Pho4$^{yEGFP}$ constitutively in the nucleus already under P$_i$-replete conditions, and *PHO5* and *PHO84* promoters were activated. These cells have strongly reduced 1,5-IP$_8$ and 5-IP$_7$, and 50% less 1-IP$_7$ than the wildtype. Thus, a decline of inositol pyrophosphates not only coincides with the induction of the P$_i$ starvation program, but the genetic ablation of these compounds is sufficient for a forced triggering of this response in P$_i$-replete conditions. Therefore, we explored the hypothesis that inositol pyrophosphates repress the PHO pathway, and that their loss upon P$_i$ starvation creates the signal that activates the starvation response.

In this case, we must explain the behavior of the *vip1Δ* mutation, which strongly reduces 1-IP$_7$ and 1,5-IP$_8$ and maintains the PHO pathway repressed in P$_i$-replete medium. Upon withdrawal of P$_i$, *vip1Δ* cells did not show nuclear relocation of Pho4 after 30 min and, even after 4 hr of starvation, *PHO5* was not expressed. The *PHO84* promoter remained partially repressed in comparison with the wildtype.

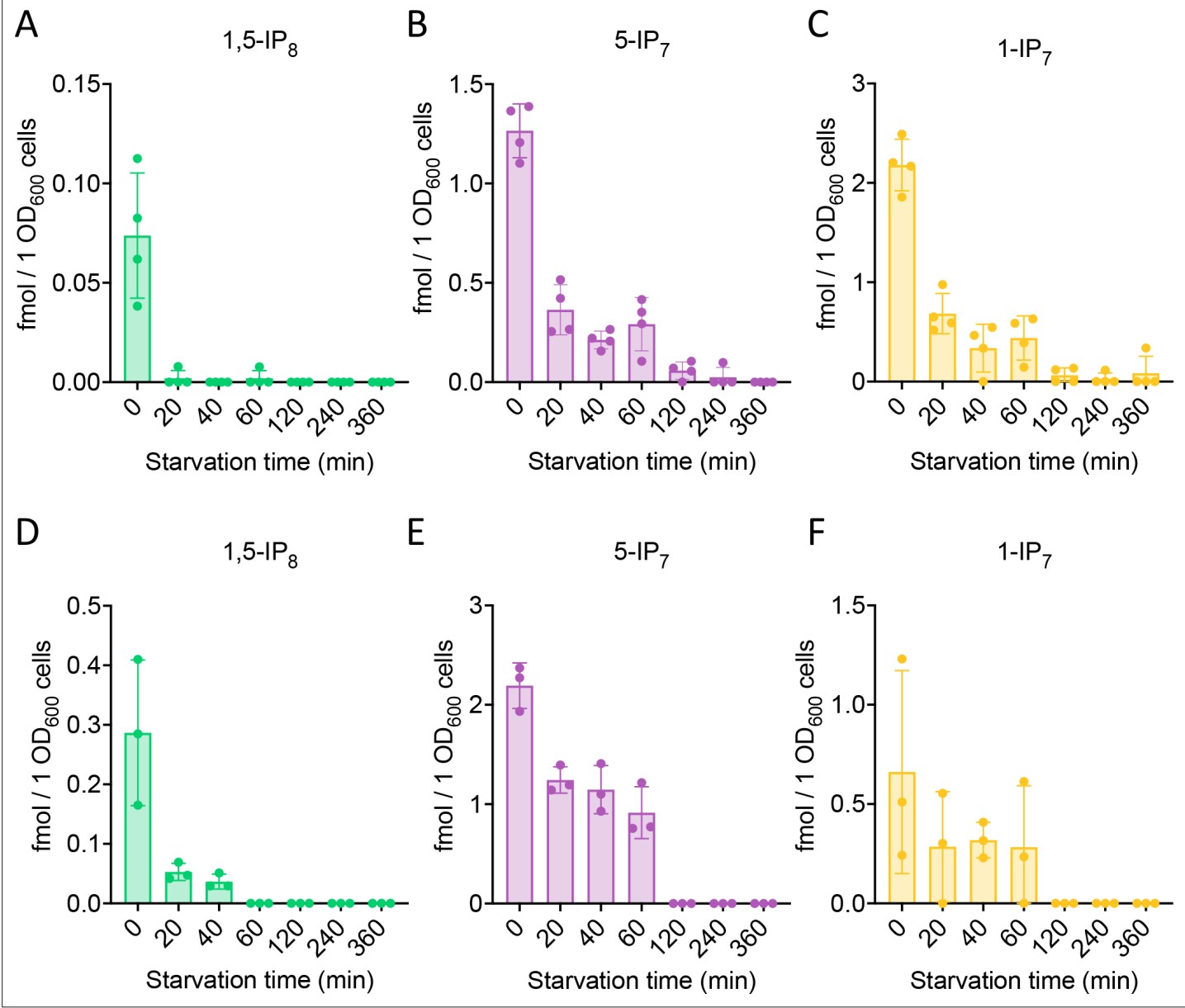

**Figure 3.** Inositol pyrophosphate analysis in *C. neoformans* and *S. pombe*. Inositol pyrophosphates were measured in *C. neoformans* (**A**) and *S. pombe* (**B**). Both fungi were logarithmically grown in synthetic complete (SC) medium for 17 h up to an $OD_{600nm}$ of 1. Cells were sedimented by centrifugation, resuspended in SC without $P_i$, and incubated further. At the indicated times, aliquots were extracted with perchloric acid. Inositol pyrophosphates were enriched on $TiO_2$ beads and analyzed by CE-MS. Concentrations of 1,5-$IP_8$, 5-$IP_7$ and 1-$IP_7$ in the extracts were determined by comparison with added synthetic $^{13}C$-labeled inositol pyrophosphate standards. The graphs provide the concentrations in the extracts. n=4 for *C. neoformans*, and n=3 for *S. pombe*; means and standard deviations are indicated. The inositol pyrophosphate values were normalized to the $OD_{600}$ of the culture for every given time point.

These results are at first sight consistent with the proposal that 1-$IP_7$ activates the PHO pathway (*Lee et al., 2007*; *Lee et al., 2008*). Several further observations draw this hypothesis into question, however. First, all three inositol pyrophosphate species strongly decline upon $P_i$ starvation instead of showing the increase postulated by Lee et al. Second, *vip1Δ* cells show a more than 15-fold overaccumulation of 5-$IP_7$. Genetic ablation of this pool by deleting *KCS1* is epistatic to the *vip1Δ* mutation. A *kcs1Δ vip1Δ* double mutant constitutively activates the PHO pathway already in presence of $P_i$, despite its strong reduction of 1-$IP_7$ and the complete absence of 1,5-$IP_8$. Third, *ddp1Δ* cells, which show a 10-fold quite selective increase in 1-$IP_7$ under $P_i$-replete conditions, did not induce the *PHO84* and *PHO5* promotors, nor did they show Pho4 accumulation in the nucleus in $P_i$-replete medium.

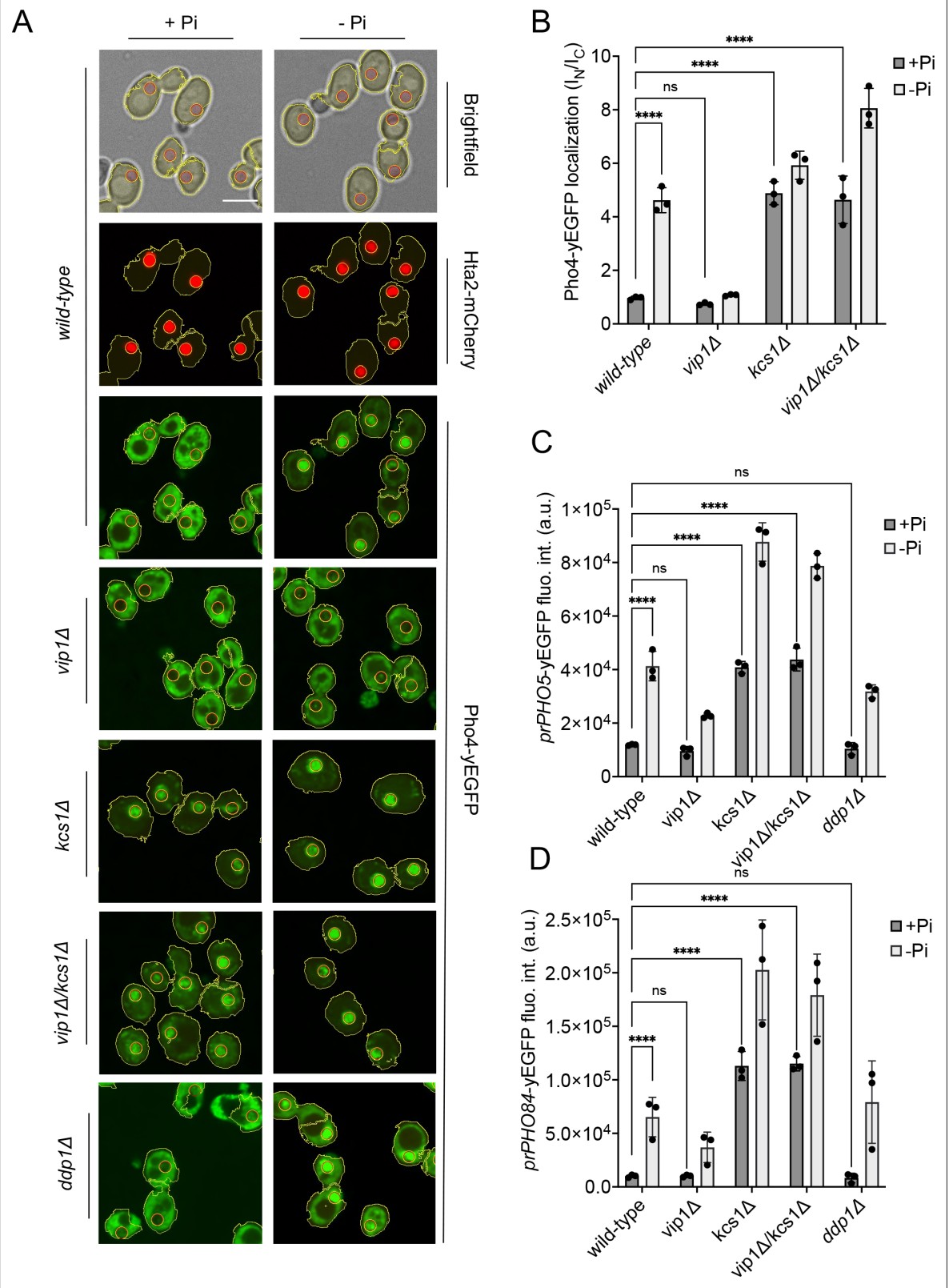

**Figure 4.** Inhibition of the PHO pathway by excessive 5-IP$_7$. The indicated cells producing Pho4$^{yEGFP}$ and the histone Hta2$^{mCherry}$ as a nuclear marker were logarithmically grown in inorganic phosphate (P$_i$)-replete synthetic complete (SC) medium, washed, and transferred to P$_i$ starvation medium as in *Figure 2A*. (**A**) Subcellular localization of Pho4$^{yEGFP}$ was analyzed on a spinning disc microscope. Cells are shown in the presence of 7.5 mM of P$_i$ (+P$_i$) or 30 min after the shift to P$_i$ starvation (- P$_i$) medium. Yellow lines surrounding the cells illustrate the segmentation performed by the algorithm that

*Figure 4 continued on next page*

Figure 4 continued

was used to quantify Pho4$^{yEGFP}$ distribution in B. Scale bar: 5 µM. $\lambda_{ex}$: 470 nm; $\lambda_{em}$: 495–560 nm. (**B**) Average intensity of Pho4$^{yEGFP}$ fluorescence was determined by automated image segmentation and analysis. Pho4$^{yEGFP}$ localization is quantified by the ratio of the average fluorescence intensities in the nucleus over the average fluorescence intensity in the cytosol ($I_N/I_C$). 100–200 cells were analyzed per condition and experiment. n=3. Means and standard deviation are indicated. (**C**) Activation of the PHO5 promotor. Cells expressing the *prPHO5-yEGFP* reporter construct from a centromeric plasmid were grown in P$_i$-replete medium (7.5 mM P$_i$) as in *Figure 2A*, and then shifted to P$_i$ starvation medium or kept in P$_i$-replete medium. After 4 hr of further incubation, fluorescence intensity of the same number of cells was measured in a Spectramax EM microplate reader. $\lambda_{ex}$: 480 nm; $\lambda_{em}$: 510 nm. n=3. Means and standard deviations are indicated. (**D**) Activation of the PHO84 promotor. Cells expressing the *prPho84-yEGFP* reporter construct from a centromeric plasmid were treated and analyzed as in C. For B, C, and D: ****p<0.0001; ***p<0.001; **p<0.01; *p<0.05; n.s. not significant, tested with Turkey's test.

The online version of this article includes the following figure supplement(s) for figure 4:

**Figure supplement 1.** Segmentation of fluorescence microscopy time-lapse experiments.

Thus, even a strong increase in 1-IP$_7$ is not sufficient to activate the PHO pathway, while reductions of inositol pyrophosphates do this.

The data described above argues against a diminution of 1-IP$_7$ as a critical factor for induction of the PHO pathway because *vip1Δ* cells, which have virtually no 1-IP$_7$, do not constitutively activate the PHO pathway. By contrast, *vip1Δ kcs1Δ* double mutants, which not only have low levels of 1-IP$_7$ and 1,5-IP$_8$ but also strongly reduced 5-IP$_7$, show constitutive PHO pathway activity (*Figure 4*). We hence tested the hypothesis that a decline of 5-IP$_7$ might trigger the PHO pathway. To define the critical concentration at which this happens in *vip1Δ* cells, we assayed the induction of the PHO pathway over time (*Figure 5*). While Pho4$^{yEGFP}$ was cytosolic in wildtype cells under high P$_i$ supply, a large fraction of it rapidly relocated into the nucleus within the first 15 min of P$_i$ starvation (*Figure 5A and B*). In *vip1Δ*, Pho4$^{yEGFP}$ relocation followed a sigmoidal curve. The nuclear/cytosolic ratio strongly increased after 1 hr, reaching a plateau after 3 hr. Thus, relocation was stimulated when the 20-fold exaggerated 5-IP$_7$ levels of the *vip1Δ* cells had declined below 5 µM (*Figure 2C*), which is six to seven times above the maximal concentration observed in P$_i$-replete wildtype cells. Since 5-IP$_7$ is the only inositol pyrophosphate that exists in *vip1Δ* cells in significant amounts, this observation places the critical concentration of 5-IP$_7$ that can repress the PHO pathway at 5 µM. In P$_i$-replete wildtype cells, 5-IP$_7$ never reaches 5 µM (*Figure 2*) and, therefore, 5-IP$_7$ is unlikely to be a physiological repressor of the PHO pathway in wildtype cells. The 5-IP$_7$-dependent PHO pathway repression in *vip1Δ* cells must then be considered as a non-physiological reaction resulting from the strong overaccumulation of 5-IP$_7$ in these cells.

This interpretation is also consistent with the phenotype of *siw14Δ* cells. Although these cells accumulate 5-IP$_7$, they accumulate it fourfold less than *vip1Δ*, remaining below the 5 µM threshold where we should expect non-physiological repression of the PHO pathway. However, *siw14Δ* cells contain the same concentration of 1,5-IP$_8$ as the wildtype (*Figure 2A*), degrade it upon P$_i$ starvation as the wildtype (*Figure 2—figure supplement 3*), and relocalize Pho4$^{yEGFP}$ with similar kinetics as the wildtype (*Figure 5B*). Taken together with the evidence presented above, 1,5-IP$_8$ thus ends up being the only inositol pyrophosphate that correlates with the activation of the PHO pathway in all mutants and conditions. This points to 1,5-IP$_8$ as the controller of the PHO pathway.

## The PHO pathway is controlled by the SPX domain of Pho81

Our results strongly argue against an increase of 1-IP$_7$ as the key factor for inactivating the CDK Pho85-Pho80 through the CDK inhibitor Pho81. This prompted us to also re-evaluate the regulation of Pho81 because previous studies had proposed a small peptide from the central region of Pho81, called the minimum domain (amino acids 645–724), as a receptor for 1-IP$_7$ that triggers the PHO pathway (*Lee et al., 2007; Lee et al., 2008*). This model rests mainly on results with the isolated minimum domain used in vitro or, in highly overexpressed form, in vivo. This expression of the minimum domain outside its normal molecular context may be problematic because earlier work had suggested that the N- and C-terminal portions of Pho81, that is regions outside the minimum domain, could provide competing inhibitory and stimulatory functions for Pho81 (*Ogawa et al., 1995*).

We targeted the N-terminal SPX domain of Pho81 through various mutations, leaving the rest of Pho81 and its minimum domain intact (*Figure 6*). First, we asked whether Pho81 lacking the N-terminal 200 amino acids, corresponding to a deletion of its SPX domain, could still activate the PHO pathway. This Pho81$^{ΔSPX}$ is likely to be folded because a yEGFP-tagged version of Pho81$^{ΔSPX}$ localized

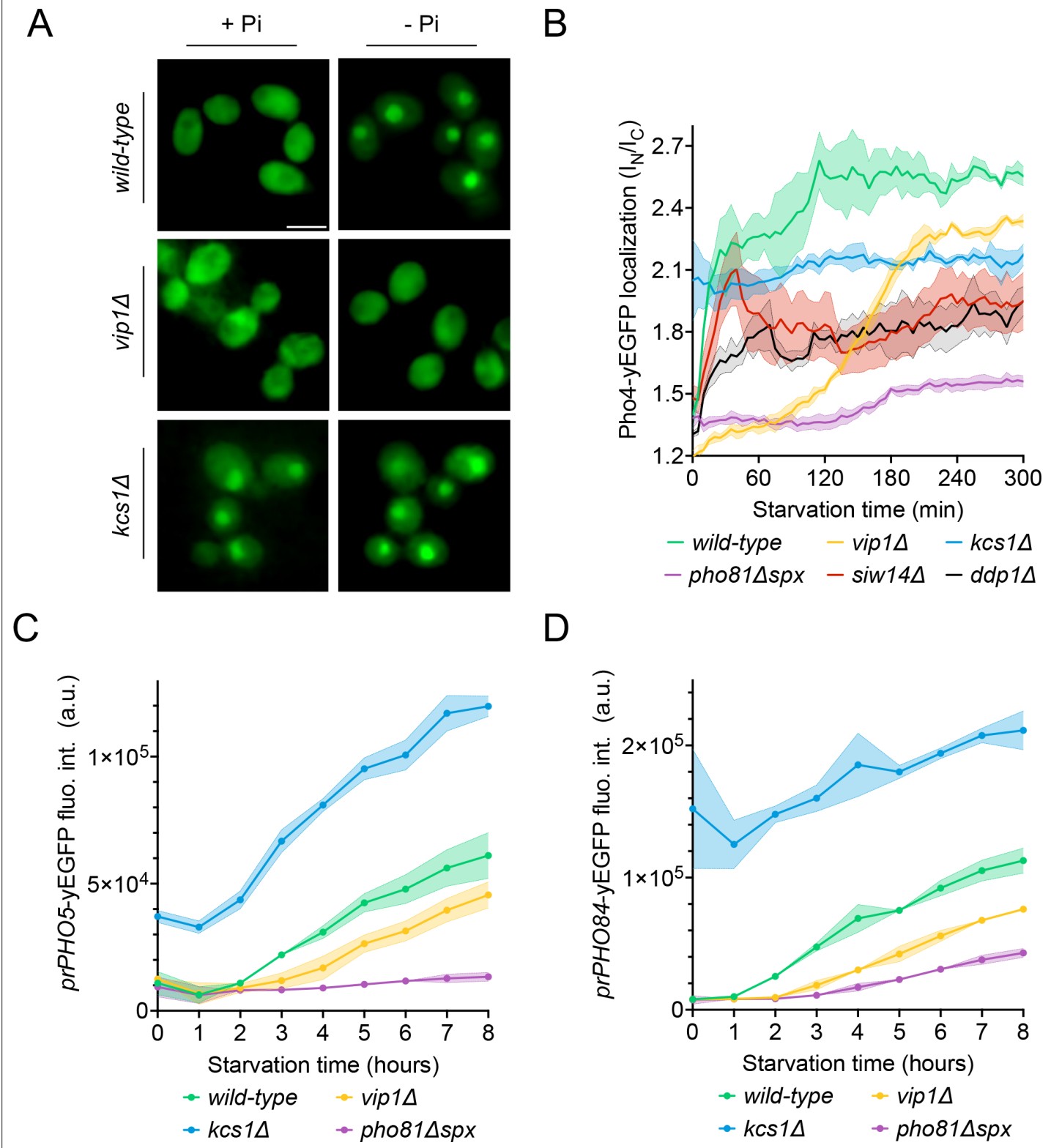

**Figure 5.** Effect of Vip1, Kcs1, and Pho81 on the time course of Pho4 translocation and PHO-gene activation. The indicated cells were logarithmically grown in inorganic phosphate ($P_i$)-replete synthetic complete (SC) medium, washed, and transferred to $P_i$ starvation medium as in *Figure 2A*. At the indicated times after transfer to Pi-starvation medium, they were analyzed for Pho4 localization, *prPHO5-yEGFP* expression, or *prPHO84-yEGFP* expression, using assays as in *Figure 4*. (**A**) Cells expressing Pho4$^{yEGFP}$ and the histone Hta2$^{mCherry}$ as a nuclear marker, shown in the presence of 7.5 mM of $P_i$ (+$P_i$) or after 90 min of $P_i$ starvation (-$P_i$). Pictures were taken on a widefield fluorescence microscope equipped with a stage-top incubator (kept

*Figure 5 continued on next page*

*Figure 5 continued*

at 30°C) and an IBIDI flow chamber. Only the GFP channel is shown. Scale bar = 5 μM. $\lambda_{ex}$: 470 nm; $\lambda_{em}$: 495–560 nm. (**B**) Distribution of Pho4[yEGFP] between the nucleus and cytosol was quantified in the cells from A at various timepoints of $P_i$ starvation. 100–200 cells were analyzed per condition and experiment at each timepoint. The solid lines and the shaded areas indicate the means and standard deviation, respectively. (**C**) Activation of the *PHO5* promotor. Cells expressing the *prPho5-yEGFP* reporter construct from a centromeric plasmid were grown in $P_i$-replete medium, shifted to $P_i$ starvation medium, and analyzed for GFP fluorescence intensity (as in *Figure 4C*) at the indicated timepoints of starvation. $\lambda_{ex}$: 480 nm; $\lambda_{em}$: 510 nm. n=3. The solid lines and the shaded areas indicate the means and standard deviation, respectively. (**D**) Activation of the *PHO84* promotor. As in C, but with cells expressing *prPHO84-yEGFP* as reporter.

to the nucleus (*Figure 6C and D*), which requires Pho81 to interact with nuclear import factors and with Pho85-Pho80 (*Huang et al., 2001*). We tested the effect of Pho81[ΔSPX] on relocation of Pho4[yEGFP] (*Figure 6A and B*). Whereas wildtype cells efficiently relocated Pho4[yEGFP] to the nucleus upon $P_i$ starvation, *pho81[ΔSPX]* cells partially maintained Pho4[yEGFP] in the cytosol. Thus, while the SPX domain is not essential for nuclear targeting of Pho81, it is required for efficient nuclear relocation of Pho4 and the induction of the PHO pathway under $P_i$ starvation.

Several lysines located on α-helix 4 of SPX domains form part of an inositol pyrophosphate-binding patch (*Ried et al., 2021*; *Pipercevic et al., 2023*). Substituting them by alanine severely reduces the affinity of the domains for inositol pyrophosphates and can thus mimic the inositol pyrophosphate-free state. We determined the corresponding lysines in the SPX domain of Pho81 (*Figure 6—figure supplement 1*), created a point mutation in the genomic PHO81 locus that substitutes one of them, K154, by alanine, and investigated the impact on the PHO pathway. Pho81[K154A-yEGFP] should be correctly folded because it concentrates in the nuclei of the cells (*Huang et al., 2001*). *pho81[K154A]* cells constitutively activated the PHO pathway because Pho4[yEGFP] accumulated in their nucleus already under $P_i$-replete conditions (*Figure 6A and B*). This constitutive activation was also observed in a *pho81[K154A] vip1Δ* double mutant, confirming that 1-IP$_7$ synthesis by Vip1 is not necessary for PHO pathway induction. Induction of the PHO pathway through SPX substitutions is further illustrated by *PHO81* expression, which itself is under the control of the PHO pathway (*Figure 6E and F*). Western blots of protein extracts from cells grown under $P_i$-replete conditions showed increased levels of Pho81 in *pho81[K154A]* cells, like those observed in strains that constitutively activate the PHO pathway due to the absence of inositol pyrophosphates, such as *kcs1Δ* and *vip1Δ/kcs1Δ*. They were significantly higher than in cells that do not constitutively activate the PHO pathway, such as wildtype or *vip1Δ*. In sum, these results suggest that the PHO pathway is regulated through binding of inositol pyrophosphates to the SPX domain of Pho81.

This critical role of the SPX domain for controlling Pho81 is also consistent with the results from two random mutagenesis approaches (*Ogawa et al., 1995*; *Creasy et al., 1993*), which had received little attention in the previous model of PHO pathway regulation (*Lee et al., 2007*; *Lee et al., 2008*). These random mutagenesis approaches generated multiple pho81[c] alleles, which constitutively activate the PHO pathway and carry the substitutions G4D, G147R, 158-SGSG-159, or E79K. All these alleles affect the SPX domain at residues in the immediate vicinity of the putative inositol pyrophosphate-binding site (*Figure 7*). Three of the alleles (G4D, G147R, 158-SGSG-159) replace small glycine residues by bulky residues or insert additional amino acids, respectively, at sites where they should sterically interfere with binding of inositol pyrophosphates. We hence expect them to mimic the inositol pyrophosphate-free state and constitutively activate the PHO pathway for this reason.

## SPX and minimum domains contribute to the interaction of Pho81 with Pho85-Pho80 in vivo

The traditional model of PHO pathway activation has tied the effect of 1-IP$_7$ to the minimum domain of Pho81, proposing that a complex of the two competitively inhibits Pho85 kinase (*Lee et al., 2008*). Since the results presented above argued against a role of 1-IP$_7$ in inhibiting Pho85-Pho80 and triggering the PHO pathway, we explored potential alternative roles of the minimum domain by modeling the Pho80-Pho81 interaction with a Google Colab notebook for Alphafold multimer v3 (*Mirdita et al., 2022*). The non-templated structure prediction of the complex (*Figure 8*) yields a Pho80 structure that agrees with an available crystal structure (*Huang et al., 2007*), showing R121 and E154 of Pho80 forming a salt bridge in a groove, which is critical for controlling Pho85 kinase (*Huang et al., 2001*). The prediction shows the groove binding a long unstructured loop of Pho81, which corresponds to

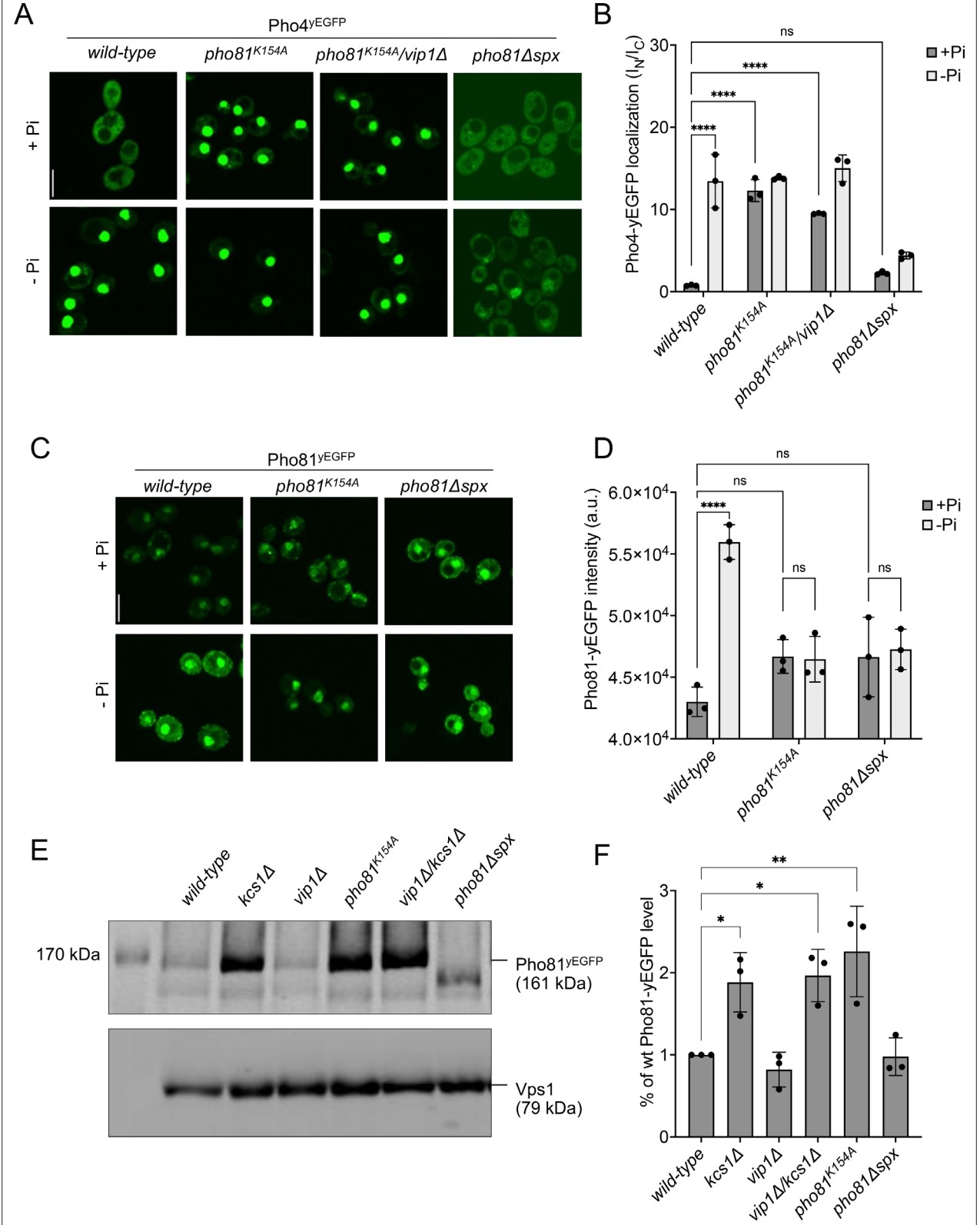

**Figure 6.** SPX-dependent activation of the PHO pathway. Cells were logarithmically grown in inorganic phosphate ($P_i$)-replete synthetic complete (SC) medium as in *Figure 2A*, washed, and incubated for further 4 hr in medium with 7.5 mM of $P_i$ (+$P_i$), or in starvation medium (- $P_i$). (**A**) Pho4 relocation. The indicated cells expressed Pho4$^{yEGFP}$ from its genomic locus. At the end of the 4 hr starvation period, GFP fluorescence was imaged on a spinning disc microscope. Scale bar: 5 µm. $\lambda_{ex}$: 488 nm; $\lambda_{em}$: 500–530 nm. (**B**) Quantification of the nuclear localization of Pho4$^{yEGFP}$ in images from A. Average

*Figure 6 continued on next page*

*Figure 6 continued*

intensity of Pho4[yEGFP] fluorescence was determined by automated image segmentation and analysis. Pho4[yEGFP] localization is quantified by the ratio of the average fluorescence intensities in the nucleus over the average fluorescence intensity in the cytosol ($I_N/I_C$). 100–200 cells were analyzed per condition and experiment. n=3. Means and standard deviation are indicated. (**C**) Pho81[yEGFP] localization. The cells expressed the indicated variants of Pho81[yEGFP] from its genomic locus. At the end of the 4 hr growth period, GFP fluorescence was imaged as in A. (**D**) Quantification of the total cellular fluorescence of Pho81[yEGFP]. Images from C were subjected to automated segmentation and the average fluorescence intensity of the entire cells was quantified as in *Figure 4B*. 100–200 cells were quantified per sample. n=3 experiments. Means and standard deviation are indicated. (**E**) Pho81[yEGFP] expression assayed by western blotting. Whole-cell protein extracts were prepared from cells expressing the indicated variants of Pho81[yEGFP], which had been grown in $P_i$-replete SC medium as in *Figure 2A*. Proteins were analyzed by SDS-PAGE and western blotting using antibodies to GFP. Vps1 was decorated as a loading control. (**F**) Quantification of Pho81[yEGFP] blotting. Bands from experiments as in E were quantified on a LICOR Odyssey infrared fluorescence imager. The signals from wildtype cells were set to 1. Means and standard deviations are shown. n=3. For B, D, and F: ****$p<0.0001$; ***$p<0.001$; **$p<0.01$; *$p<0.05$; n.s. not significant, determined with Turkey's test.

The online version of this article includes the following figure supplement(s) for figure 6:

**Figure supplement 1.** *Myo*-inositol polyphosphate-binding pocket of yeast SPX domains.

the minimum domain. Substitutions in the minimum domain, and specifically of the residues binding the Pho80 groove in the prediction (residues 690–701), constitutively repress the PHO pathway when introduced into full-length Pho81, and they destabilize the interaction of Pho81 with Pho85-Pho80 (*Ogawa et al., 1995*; *Huang et al., 2001*). The same effects are also shown by the *pho80[R121K]* and *pho80[E154V]* alleles, which ablate the critical salt bridge in the Pho80 groove (*Huang et al., 2001*; *Huang et al., 2007*). The similarity of the effects of these substitutions validates the predicted interaction of the minimum domain with the R121/E154-containing groove. It invites a re-interpretation of the role of the minimum domain and suggests that this domain may serve as an anchor for Pho80 on Pho81 rather than acting directly on the Pho85 kinase.

The recruitment of Pho81 to the nucleus and its interaction with Pho85-Pho80 in wildtype cells is independent of the $P_i$ regime, suggesting that it is dominated by a constitutive interaction (*Figure 6C*; *Huang et al., 2001*). This constitutive recruitment vanishes when the minimum domain-binding site is compromised in *pho80[R121K]* and *pho80[E154V]* cells. Then, recruitment of Pho81 to the nucleus and to Pho85-Pho80 becomes $P_i$-sensitive and is stimulated by $P_i$ starvation (*Figure 9*). We hence used *pho80[R121K]* and *pho80[E154V]* as tools to ask whether Pho81 might be tied to Pho85-Pho80 through a

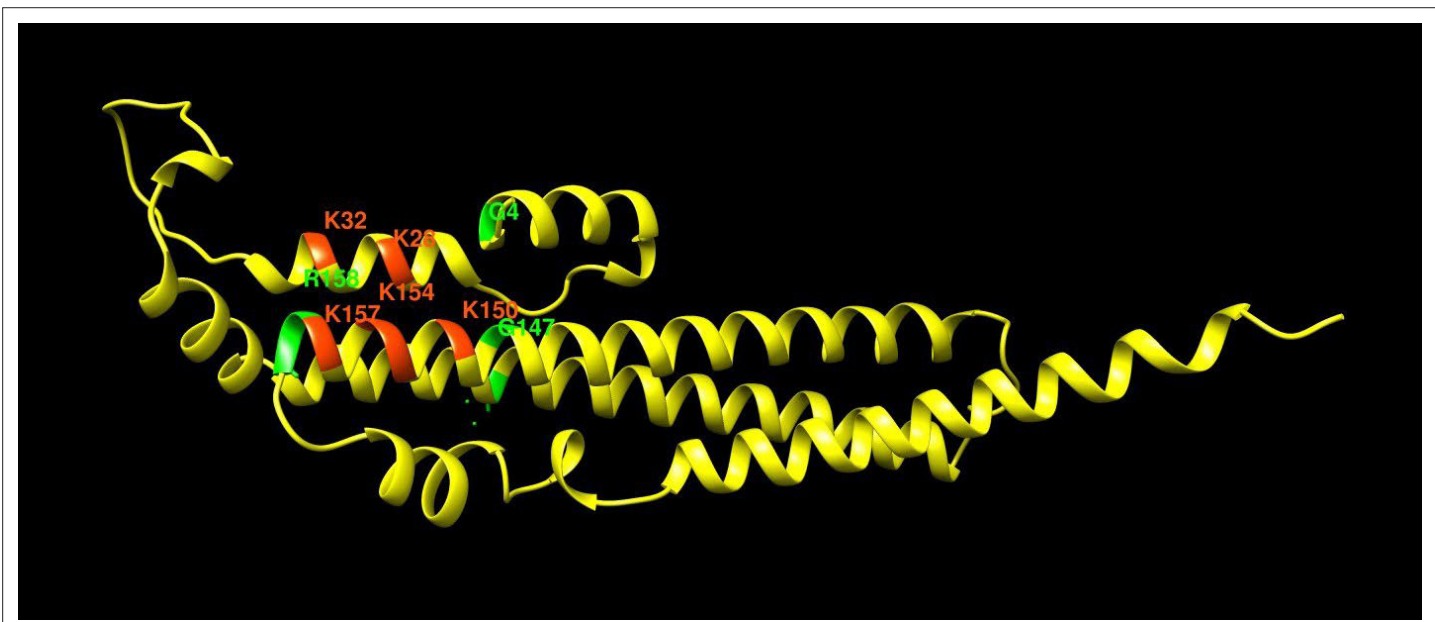

**Figure 7.** Pho81 residues leading to constitutive activation of the PHO pathway. The image shows an Alphafold prediction of the Pho81 SPX domain (amino acids 1–215), taken from Alphafold database model AF-P17442-F1 (*Varadi et al., 2022*), in yellow. Basic residues of the putative inositol pyrophosphate-binding site have been identified by structure matching with the IP$_6$-associated SPX domain of *VTC4* from *C. thermophilum* (PDB 5IJP). They are labeled in red. Residues from random mutagenesis screens, which lead to constitutive activation of the PHO pathway, are labeled in green.

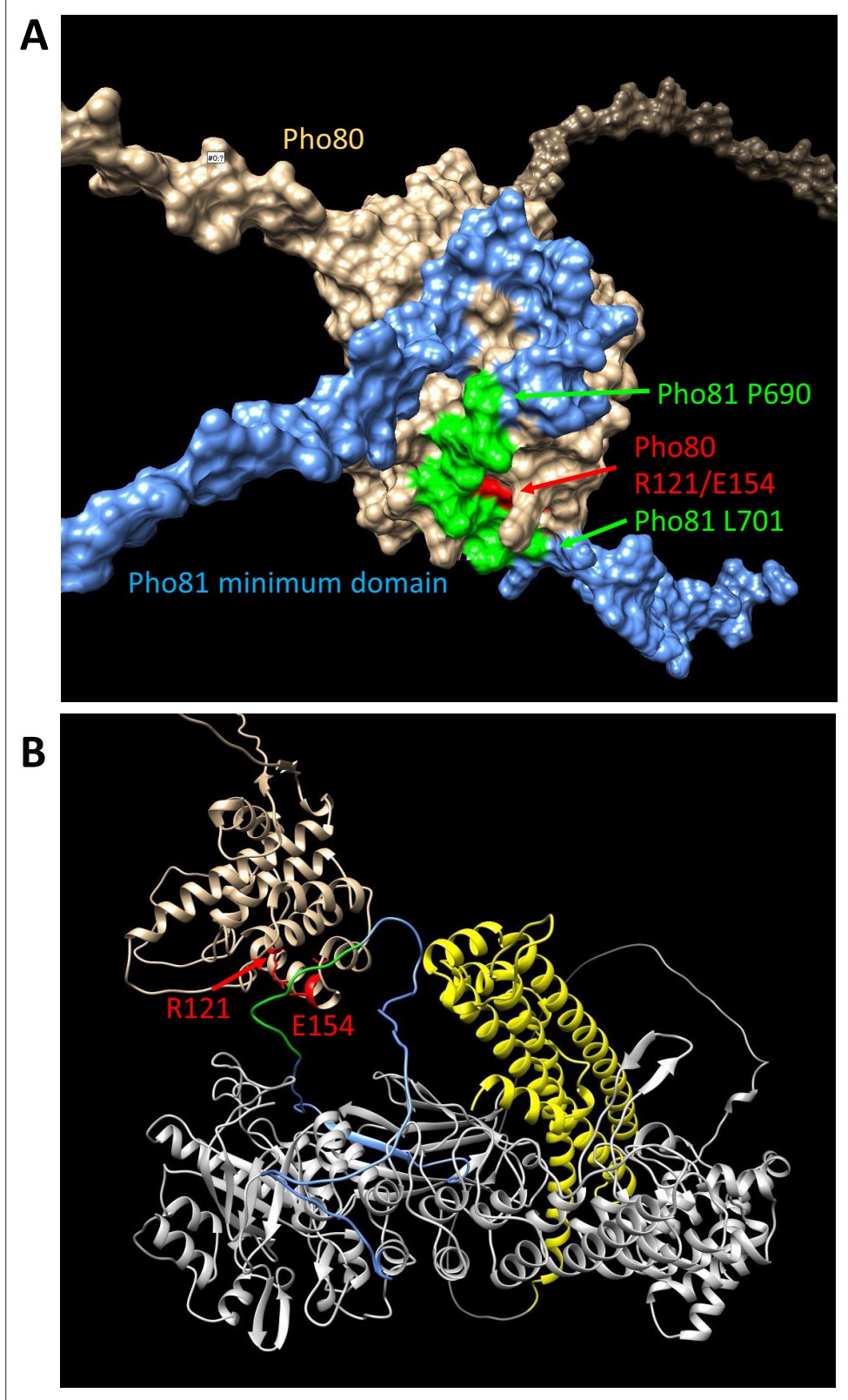

**Figure 8.** Structure predictions of the minimum domain in complex with Pho80. Alphafold multimer v3 was used to generate the following structure predictions. (**A**) Surface representation of Pho80 in complex with a peptide corresponding to the minimum domain of Pho81 (residues 645–724), showing the embedding of critical residues (690–701) of the minimum domain in a groove of Pho80 that contains the salt bridge between R121 and E154.

*Figure 8 continued on next page*

*Figure 8 continued*

(**B**) Ribbon representation of Pho80 in complex with full-length Pho81. Coloring: Beige - Pho80; red - E154, R121 of Pho80; blue: Pho81 minimum domain; green - central region of the Pho81 minimum domain that is critical for Pho85-Pho80 inhibition (aa 690–701); yellow: SPX domain of Pho81; gray - rest of Pho81 without assigned function. The complete dataset for both predictions has been deposited at Figshare under the DOI 10.6084 /m9. figshare.c.6700281.

second layer of interaction that provides $P_i$ dependence. As a bona fide $P_i$-responsive element, the SPX domain of Pho81 was a prime candidate. We combined *pho80$^{R121K}$* and *pho80$^{E154V}$* with substitutions in the putative inositol pyrophosphate-binding site of a fluorescent Pho81$^{yEGFP}$ fusion protein, and with deletions of VIP1 or KCS1. Under $P_i$-replete conditions, *pho80$^{R121K}$* or *pho80$^{E154V}$* cells showed Pho81$^{yEGFP}$ in the cytosol, but they accumulated the protein in the nucleus upon $P_i$ starvation (**Figure 9**). This rescue by starvation was more pronounced in *pho80$^{R121K}$* than in *pho80$^{E154V}$*. Pho81$^{K154A-yEGFP}$, with a substitution in its putative inositol pyrophosphate binding site designed to mimic the low-$P_i$ state, accumulated in the nucleus of *pho80$^{R121K}$* and *pho80$^{E154V}$* cells already under $P_i$-replete conditions, that is, it behaved constitutively as under $P_i$ starvation. *vip1Δ* mutants behaved like the wildtype, accumulating Pho81$^{yEGFP}$ in the nucleus *of pho80$^{R121K}$* and *pho80$^{E154V}$* cells upon $P_i$ starvation, whereas *kcs1Δ* cells showed nuclear Pho81$^{yEGFP}$ already under $P_i$-replete conditions. Thus, interfering with inositol pyrophosphate signaling, be it through $P_i$ starvation, ablation of inositol pyrophosphate synthesis, or substitution of the putative inositol pyrophosphate binding site of Pho81, partially compensated for the destabilization of the minimum domain binding site in Pho80.

Two in vivo observations are consistent with the notion that this compensation may reflect an interaction of the SPX domain with Pho85-Pho80: First, Pho81$^{ΔSPX-yEGFP}$, which lacks this SPX domain, did not show starvation-induced nuclear accumulation in *pho80$^{R121K}$* and *pho80$^{E154V}$* cells (**Figure 9**). Second, a yEGFP fusion of only the SPX domain of Pho81 (Pho81SPX$^{yEGFP}$), which lacks the rest of the protein and thereby excludes a constitutive interaction through the minimum domain, localized mainly to the cytosol of PHO80 wildtype cells under $P_i$-replete conditions, but it concentrated in the nucleus upon $P_i$ starvation (**Figure 10**). This concentration was not observed in *pho80Δ* cells, which lack nuclear Pho85-Pho80. By contrast, *pho80$^{R121K}$* or *pho80E$^{154V}$* cells, which only target the minimum domain-Pho80 interaction but leave Pho85-Pho80 kinase active and in the nucleus (**Huang et al., 2007**), retained the capacity to accumulate Pho81SPX$^{yEGFP}$ in their nuclei upon $P_i$ starvation. These results suggest that the SPX domain contributes to the interaction of Pho81 and Pho85-Pho80. Our observations are consistent with a model in which this SPX-Pho85-Pho80 interaction is independent of the minimum domain but controlled by $P_i$ through inositol pyrophosphates, which can destabilize it and relieve the inhibition of Pho85-Pho80.

## Discussion

The transcriptional response of *S. cerevisiae* to varying $P_i$ supply is an important model system for studying gene regulation, promotor structure, and the impact of chromatin structure (**Korber and Barbaric, 2014**; **Wolff et al., 2021**). However, identification of the inositol pyrophosphates relaying phosphate availability to this PHO pathway has been challenging. This is mainly due to limitations in analytics. The recent development of a CE-MS method has provided a potent tool, allowing inositol pyrophosphate analysis without radiolabeling and under any physiological condition. The improved separation, higher throughput, and lower cost of this approach led us to re-investigate the role of inositol pyrophosphates in PHO pathway regulation. The results inspire a revised model that differs from the traditional hypothesis in three aspects: We propose that $P_i$ scarcity is signaled by a decrease in inositol pyrophosphates rather than by an increase; a critical function is ascribed to the loss of 1,5-IP$_8$ instead of 1-IP$_7$; and we postulate that the N-terminal SPX domain of Pho81 rather than the central minimum domain is the primary receptor mediating inositol pyrophosphate control over the PHO pathway.

Our observations reveal an effect that misled the interpretation of several preceding studies. This includes our own work on the activation of the polyphosphate polymerase VTC (**Hothorn et al., 2009**), which is controlled by SPX domains (**Gerasimaite et al., 2017**; **Wild et al., 2016**; **Pipercevic et al., 2023**; **Gerasimaitė and Mayer, 2016**). Our in vitro activity assays had shown a clear order of potency

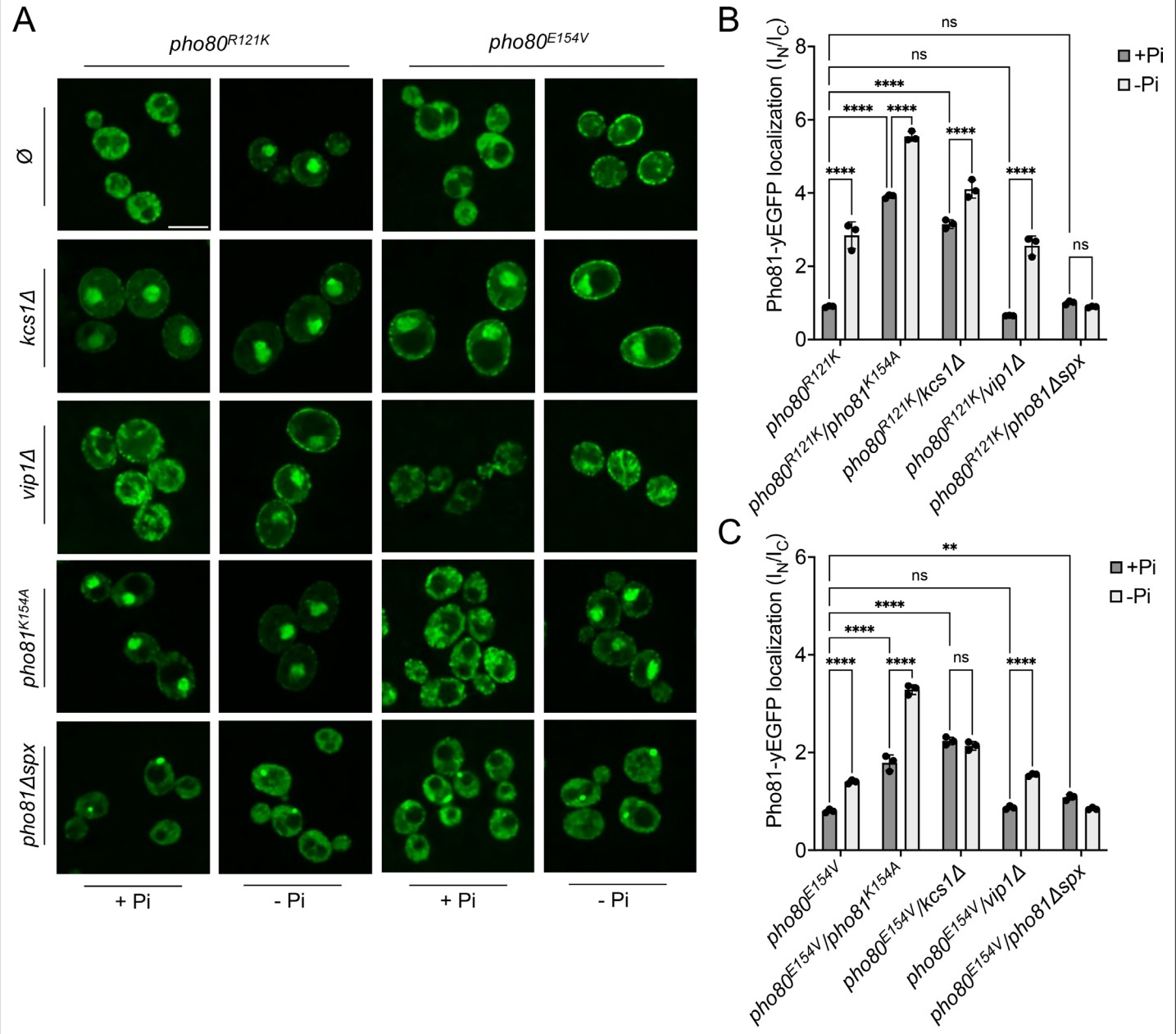

**Figure 9.** Localization of Pho81$^{yEGFP}$ in $pho80^{R121K}$ and $pho80^{E154V}$ loss-of-affinity mutants. Cells were logarithmically grown in inorganic phosphate (P$_i$)-replete synthetic complete (SC) medium as in **Figure 2A**, washed, and incubated for further 4 hr in medium with 7.5 mM of P$_i$ (+P$_i$), or in starvation medium (-P$_i$). (**A**) Pho81 imaging. Cells expressing the indicated variants of Pho81$^{yEGFP}$ and Pho80 from their genomic loci were imaged on a spinning disc microscope after 4 hr of growth in the presence of 7.5 mM Pi (+P$_i$), or after 4 hr of P$_i$ starvation (- P$_i$). Scale bar = 5 μM. $\lambda_{ex}$: 488 nm; $\lambda_{em}$: 500–530 nm. Note that the fluorescent dots visible in $pho81^{\Delta spx}$ cells are not nuclei because they are too small and at positions where nuclei are not found. Their location was not further investigated because it is not essential for this study. (**B**) Quantification of the nuclear localization of Pho81$^{yEGFP}$ in $pho80^{R121K}$ cells. Images from A were subjected to automated segmentation and quantification of the average fluorescence intensity in the nucleus and cytosol as in **Figure 4B**. 100–200 cells were quantified per sample. n=3 experiments. (**C**) Quantification of the nuclear localization of Pho81$^{yEGFP}$ in $pho80^{E154V}$ mutants was performed as in B. For B and C: ****p<0.0001; ***p<0.001; **p<0.01; *p<0.05; n.s. not significant, determined with Turkey's test.

of inositol pyrophosphates in stimulating VTC, with 1,5-IP$_8$ showing an apparent EC$_{50}$ 30-fold below that of 5-IP$_7$ and 50-fold below that of 1-IP$_7$ (**Gerasimaite et al., 2017**). In this work, we nevertheless dismissed 1,5-IP$_8$ as a physiological stimulator of the VTC SPX domains, based on the argument that a $vip1\Delta$ mutant, which lacks the enzyme for 1-IP$_7$ and 1,5-IP$_8$ synthesis, shows quite normal polyP accumulation in vivo. We proposed 5-IP$_7$ for this role because a $kcs1\Delta$ mutant, which lacks the IP6K making

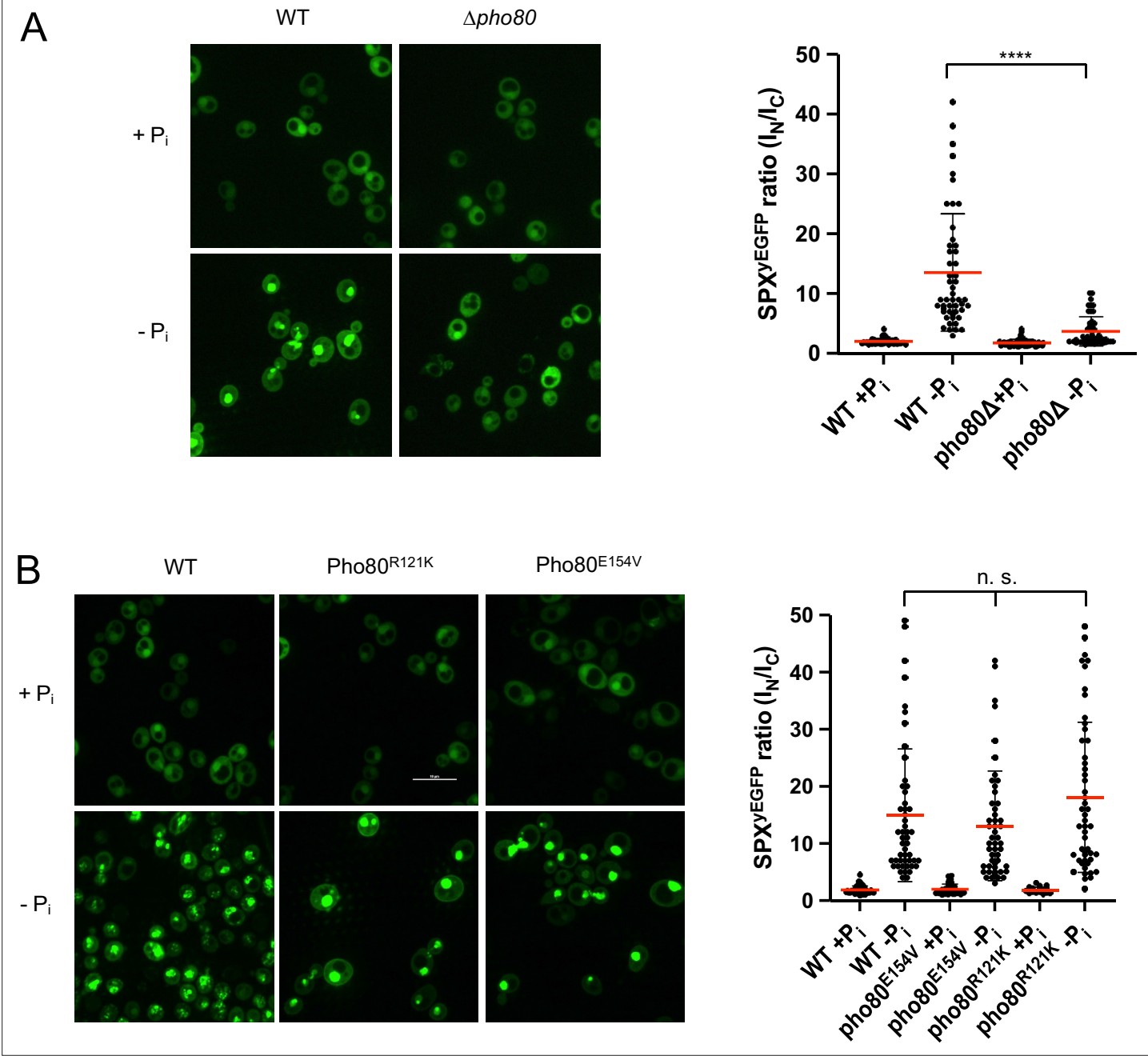

**Figure 10.** Recruitment of Pho81SPX$^{yEGFP}$ to the nucleus by Pho85-Pho80. The indicated wildtype or isogenic mutant cells expressing the SPX domain of Pho81 as a yEGFP fusion (Pho81SPX$^{yEGFP}$) from a centromeric plasmid under the ADH promotor were logarithmically grown in synthetic complete (SC), sedimented in a tabletop centrifuge, and transferred into SC with (+P$_i$) or without P$_i$ (-P$_i$). After 3 hr of further cultivation, cells were imaged by spinning disc confocal microscopy. 50 cells per condition were analyzed from two independent experiments. Regions of interest were defined manually, and the fluorescence contained in the nuclei (I$_N$) or the cytosol (I$_C$) was integrated using ImageJ. Scale bar: 5 µm. Indicated pairwise differences were evaluated by a Mann-Whitney test. ****p<0.0001; n.s. not significant. (**A**) WT (BY4742) and isogenic *pho80Δ* cells expressing *PHO81SPX$^{yEGFP}$*. (**B**) WT (BY4741) and isogenic cells expressing *pho80$^{R121K}$* or *pho80$^{E154V}$* from their genomic locus and Pho81SPX$^{yEGFP}$ from the plasmid.

this compound, also lacks polyP. The inositol pyrophosphate analyses that we present now call for a re-interpretation of this data. The presence of polyP in a *vip1Δ* mutant can well be explained by the 20-fold overaccumulation of 5-IP$_7$ in this mutant, which can compensate for the lower EC$_{50}$ and permit stimulation of VTC through 5-IP$_7$. In wildtype cells, however, where 5-IP$_7$ is only twice as abundant as

1,5-IP$_8$ (*Figure 2*), 1,5-IP$_8$ should be the relevant activator of VTC for polyP synthesis under P$_i$-replete conditions.

Then, the situation is also compatible with results from *S. pombe*, where ablation of the Vip1 homolog Asp1 impairs polyP synthesis (*Pascual-Ortiz et al., 2021*; *Schwer et al., 2022*). The transcriptional phosphate starvation response in *S. pombe* also depends on inositol pyrophosphates. However, the downstream executers differ from those present in *S. cerevisiae*. Fission yeast contains no homologs of Pho81, and the responsible transcription factor is not controlled through a Pho85-Pho80-like kinase (*Estill et al., 2015*; *Henry et al., 2011*; *Carter-O'Connell et al., 2012*). P$_i$-dependent transcription strongly depends on lncRNA-mediated interference, which may be linked to inositol pyrophosphates through an SPX domain containing ubiquitin E3 ligase (*Sanchez et al., 2019*; *Schwer et al., 2021*; *Sanchez et al., 2018*; *Schwer et al., 2022*). Nevertheless, important basic properties of the IP6K and PPIP5K system are conserved. Like in baker's yeast, deletion of the Vip1 homolog Asp1 leads to a strong increase in IP$_7$ (*Pascual-Ortiz et al., 2018*). Characterization of the purified enzyme also showed that P$_i$ stimulates net production of IP$_8$ by inhibiting the phosphatase domain of Asp1 (*Benjamin et al., 2022*). In line with this, we observed that IP$_8$ rapidly decreased upon P$_i$ starvation also in *S. pombe* (*Figure 3*). As for Vip1 in baker's yeast, genetic ablation of Asp1 does not interfere with the induction of the phosphate-regulated gene *PHO1* by P$_i$ starvation, suggesting that 1-IP$_7$ and 1,5-IP$_8$ are not essential for this transcriptional response (*Estill et al., 2015*). In P$_i$-replete media, however, ablation of Asp1 hyper-represses *PHO1*, and putative overproduction of 1-IP$_7$ and/or 1,5-IP$_8$ by selective inactivation of only the Asp1 phosphatase domain induces this gene (*Estill et al., 2015*; *Garg et al., 2020*). It remains to be seen whether these discrepancies originate from a similar disequilibration of IP$_7$ species as in *S. cerevisiae* (ablation of Asp1 increases the IP$_7$ pool four- to fivefold *Pascual-Ortiz et al., 2018*), or whether they are due to other processes such as lncRNA-mediated interference, which plays an important role in P$_i$ homeostasis in *S. pombe* (*Sanchez et al., 2018*; *Schwer et al., 2015*; *Schwer et al., 2014*).

The PHO pathway in *C. neoformans*, a pathogenic yeast, shares many protein components with that of *S. cerevisiae* and might hence be expected to function in a similar manner (*Toh-e et al., 2015*). Yet a recent study on this organism proposed 5-IP$_7$ as the critical inducer of the PHO pathway (*Desmarini et al., 2020*). This model relies on the observation that *kcs1Δ* mutants, which lack 5-IP$_7$, cannot induce the PHO pathway in this yeast. This is the opposite behavior compared to *S. cerevisiae*. It was hence concluded that, despite an experimentally observed moderate decline in 5-IP$_7$ upon short P$_i$ starvation, the remaining pool of 5-IP$_7$ was required for PHO pathway activity. Substitution of basic residues in the inositol pyrophosphate-binding pocket of *C. neoformans* Pho81 interfered with PHO pathway activation, lending further support to the conclusion. However, the same study also showed that Pho81 was lacking completely from kcs1Δ mutants. When the Pho81 inositol pyrophosphate-binding site was compromised by amino acid substitutions, the Pho81 protein level was significantly reduced, and the protein almost vanished upon P$_i$ starvation. This offers an alternative explanation of the effects because absence of the critical inhibitor Pho81 activates Pho85/80 kinase and represses the PHO pathway independently of P$_i$ availability (*Schneider et al., 1994*; *Ogawa et al., 1995*). Destabilization of Pho81 will then mask effects of inositol pyrophosphates. Thus, it remains possible that the PHO pathway in *C. neoformans* is controlled as in *S. cerevisiae*, that is that P$_i$ starvation is signaled by a decline of 1,5-IP$_8$. In line with this, we observed that *C. neoformans* and *S. pombe* cells lose all inositol pyrophosphates within 2 hr of phosphate starvation (*Figure 3*). As in *S. cerevisiae*, 1,5-IP$_8$ showed a faster and more profound response. Therefore, we consider it as likely that the loss of 1,5-IP$_8$ signals P$_i$ starvation also in *C. neoformans* and *S. pombe*.

Our results revise the widely accepted model of increased 1-IP$_7$ as the trigger of the PHO pathway and of the minimum domain of Pho81 as its receptor (*Lee et al., 2007*; *Lee et al., 2008*). This model rests upon an increase of 1-IP$_7$ upon P$_i$ starvation, a reported constitutive activation of the PHO pathway in a *ddp1Δ* mutant, and upon the lack of rapid PHO pathway induction in a *vip1Δ* mutant. A later study from O'Shea and colleagues revealed that *vip1Δ* cells finally induced the PHO pathway after prolonged P$_i$ starvation (*Choi et al., 2017*), but this did not lead to a revision of the model. Our results rationalize the strong delay of *vip1Δ* in PHO pathway induction by the overaccumulation of 5-IP$_7$ in this mutant, which leads to aberrant PHO pathway silencing through this compound. Under P$_i$ starvation, it takes several hours longer to degrade this exaggerated pool, which provides a satisfying explanation for the delayed PHO pathway induction in *vip1Δ* cells. Even though earlier studies, due to

the limited resolution of the radio-HPLC approach, could not differentiate between 1- and 5-IP$_7$, these studies reported an increase in the IP$_7$ pool upon P$_i$ starvation (*Lee et al., 2007*). Other studies using a similar radio-HPLC approach suggested a decline of the IP$_7$ pool upon long P$_i$ withdrawal, but they analyzed only a single timepoint and could not exclude an earlier increase in 1-IP$_7$ as a trigger for the PHO pathway (*Wild et al., 2016*; *Desmarini et al., 2020*; *Lonetti et al., 2011*). In our CE-MS analyses, we consistently observed strong, continuous declines of 1,5-IP$_8$, 5-IP$_7$, and 1-IP$_7$ and, upon several hours of P$_i$ starvation, their almost complete disappearance. Furthermore, we could not confirm the reported constitutive activation of the PHO pathway in a *ddp1Δ* mutant (*Lee et al., 2007*), although this mutant did show the expected increased 1-IP7 concentration. These observations argue against a role of 1-IP$_7$, 5-IP$_7$, or 1,5-IP$_8$ as activators for Pho81 and the PHO pathway.

The link between 1-IP$_7$ and Pho81 activation relied mainly on the use of the 80 amino acid minimum domain from the central region (*Lee et al., 2007*; *Lee et al., 2008*; *Huang et al., 2001*). This domain was used in in vitro experiments, where 1-IP$_7$ stimulated binding of the minimum domain to Pho85-Pho80 and inhibition of the enzyme. Given that 1-IP$_7$ disappears under P$_i$ starvation, it appears unlikely that this in vitro effect is physiologically relevant. Furthermore, the apparent dissociation constant of the complex of 1-IP$_7$ with the minimum domain and Pho80-Pho85 was determined to be 20 µM (*Lee et al., 2008*). The minimum domain is thus unlikely to respond to 1-IP$_7$ in the cells because its $K_d$ is almost 50-fold above the cellular concentrations of 1-IP$_7$ in P$_i$-replete medium (0.5 µM; *Figure 2*), and at least three orders of magnitude above the concentration remaining on P$_i$ starvation media (<20 nM). Another discrepancy is that the in vitro inhibition of Pho85-Pho80 by the minimum domain was reported to be specific for 1-IP$_7$ whereas 5-IP$_7$ had no effect (*Lee et al., 2008*). This contrasts with the in vivo situation, where we find that even massive overaccumulation of 1-IP$_7$ by ablation of DDP1 cannot trigger the PHO pathway, but where suppression of 5-IP$_7$ and 1,5-IP$_8$ production by ablation of KCS1 does activate it constitutively (*Figure 4*; *Auesukaree et al., 2005*).

An argument supporting a role of the minimum domain in mediating P$_i$ responsiveness was provided by in vivo experiments in a *pho81Δ* mutant, where P$_i$-dependent activation of the PHO pathway could be partially rescued through overexpression of the minimum domain (*Ogawa et al., 1995*; *Huang et al., 2001*). PHO pathway activation was assayed through *PHO5* expression. This is relevant because

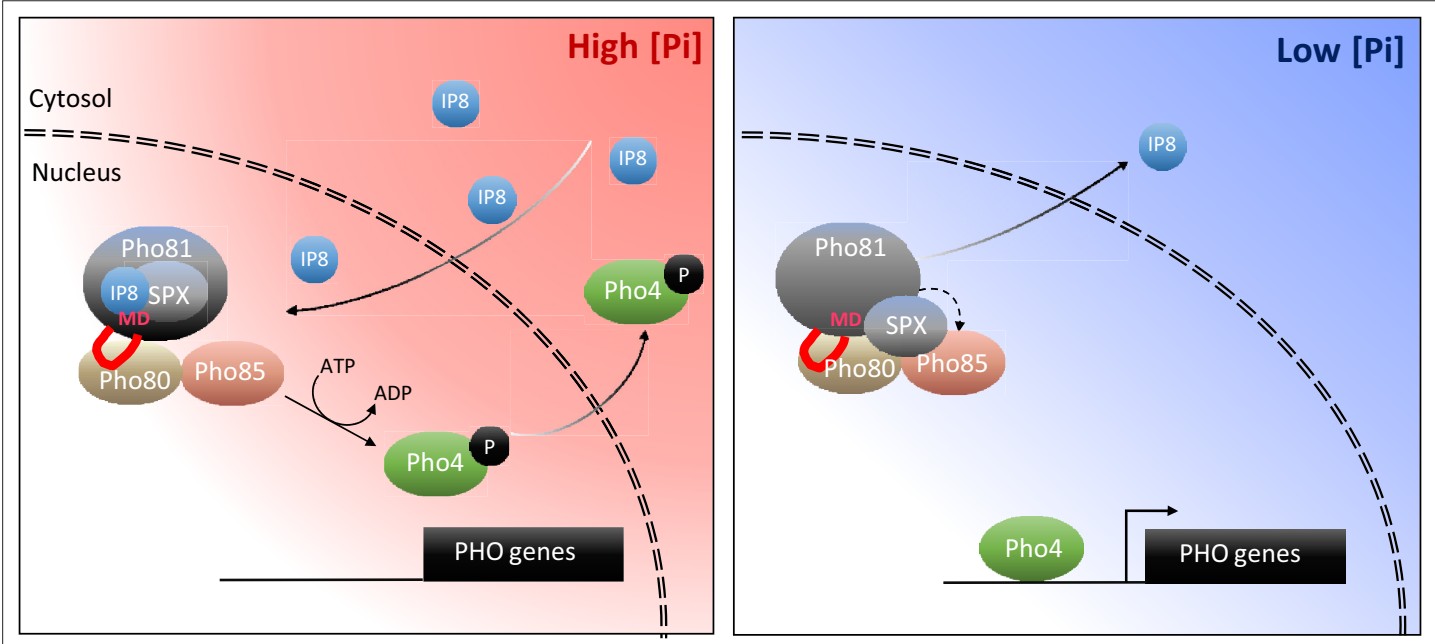

**Figure 11.** Working model on the control of the PHO pathway through 1,5-IP$_8$ and Pho81. At high P$_i$ concentrations, inositol pyrophosphates accumulate. 1,5-IP$_8$ binds the SPX domain of Pho81, which labilizes the interaction of Pho81 with Pho80 and prevents Pho81 from inhibiting Pho85-Pho80 kinase. In low P$_i$, 1,5-IP$_8$ declines. Liberation of the SPX domain from this ligand allows this domain to interact with Pho85-Pho80. This, and the interaction of the minimum domain with a critical groove of Pho80, allow Pho81 to inhibit Pho80-85. The resulting dephosphorylation of Pho4 triggers its concentration in the nucleus and activation of the PHO pathway. This inhibition is reinforced by increased expression of PHO81, which itself is a PHO pathway-controlled gene.

the *PHO5* promotor is not only controlled through Pho85-Pho80-mediated phosphorylation of Pho4, but also through changes in nucleosome occupancy, which substantially influence activity and activation threshold of the *PHO5* promotor (*Almer et al., 1986*; *Lam et al., 2008*). It is thus conceivable that constitutive overexpression of the minimum domain reduces Pho85-Pho80 activity moderately, but sufficiently to allow regulation of *PHO5* expression through nucleosome positioning, which changes in a $P_i$-dependent manner (*Almer et al., 1986*).

Even though the minimum domain is unlikely to function as a receptor for inositol pyrophosphates, this does not call into question that the minimum domain is critical for Pho81 function. A role of this domain is supported by substitutions and deletions in this domain, introduced into full-length Pho81, which constitutively repress the PHO pathway (*Ogawa et al., 1995*; *Huang et al., 2001*). Our modeling approach provides a satisfying link of these residues to Pho85-Pho80 regulation. It suggests the minimum domain to be an extended unstructured loop that binds Pho80 at a groove that encompasses R121 and E154, residues that emerged from a genetic screen as essential for recruiting Pho81 to Pho85-Pho80 and for inhibiting the kinase (*Huang et al., 2001*). Therefore, we propose that the minimum domain does provide a critical connection between Pho81 and Pho80, which is per se inositol pyrophosphate-independent, whereas the SPX domain of Pho81 may create additional contacts to Pho80-Pho85 and impart regulation through inositol pyrophosphates.

Taking all these aspects together, we propose the following working model (*Figure 11*): $P_i$ sufficiency in the cytosol is signaled through the accumulation of $1,5\text{-IP}_8$. $1,5\text{-IP}_8$ binding to the SPX domain of Pho81 labilizes the interaction of this domain with Pho85-Pho80, allowing the kinase to become active. The now un-inhibited kinase phosphorylates Pho4, triggers its relocation into the cytosol and represses the PHO pathway. $P_i$ limitation is signaled by a decline in $1,5\text{-IP}_8$, resulting in the ligand-free form of the SPX domain and the activation of Pho81. Inhibition of Pho85-Pho80 requires the Pho81 minimum domain to bind a groove in the cyclin Pho80, stabilizing the complex and inhibiting Pho85 kinase. An obvious question arising from this model is whether the Pho81 SPX domain itself could activate Pho85-Pho80 by directly binding these proteins. Alternatively, or in addition, Pho85-Pho80 might be controlled through the availability of the minimum domain for Pho80 binding, which might be restricted through an inositol pyrophosphate-dependent SPX-minimum domain interaction. These aspects will be explored in future work.

This revision of PHO pathway regulation permits to develop a coherent picture on intracellular phosphate signaling across the eukaryotic kingdoms. Phosphate abundance being indicated by the accumulation of $1,5\text{-IP}_8$ is compatible with the situation in mammalian cells, where $1,5\text{-IP}_8$ triggers $P_i$ export through the putative phosphate transporter XPR1 (*Li et al., 2020*; *Giovannini et al., 2013*), and with the situation in plants, where $IP_8$ is particularly abundant upon shift to high-$P_i$ substrate and inhibits transcription factors for the phosphate starvation response (*Ried et al., 2021*; *Dong et al., 2019*; *Riemer et al., 2021*; *Zhu et al., 2019*). Thus, loss of $1,5\text{-IP}_8$ may generally signal phosphate scarcity in eukaryotic cells.

**Table 1.** Parameters for MRM transitions.

| Compound | Precursor ion | Product ion | dwell | CE (V) | Cell acc (V) | Polarity |
|---|---|---|---|---|---|---|
| $[^{13}C_6]IP_8$ | 411.9 | 362.9 | 80 | 10 | 1 | Negative |
| $IP_8$ | 408.9 | 359.9 | 80 | 10 | 1 | Negative |
| $[^{13}C_6]IP_7$ | 371.9 | 322.9 | 80 | 10 | 1 | Negative |
| $IP_7$ | 368.9 | 319.9 | 80 | 10 | 1 | Negative |
| $[^{13}C_6]IP_6$ | 331.9 | 486.9 | 80 | 17 | 1 | Negative |
| $IP_6$ | 328.9 | 480.9 | 80 | 17 | 1 | Negative |

# Materials and methods

## Materials and data availability

All strains and plasmids used in this study (listed in *Supplementary file 1*) are available from the corresponding author upon request. Software and datasets are available from Figshare under the DOI 10.6084 /m9.figshare.c.6700281.

## Strains

Detailed lists of strains, plasmids, and primers used in this study are provided in the appendix.

The *S. cerevisiae* strains used in this study were all obtained by genetic manipulations from the BY4741 background described in *Table 1*. This strain corresponds to the wildtype of this study. For the sake of clarity, the BY4741 background was therefore not indicated in the genotype of each mutant strain.

## Yeast culture

Unless stated otherwise, all experiments were performed with *S. cerevisiae* BY4741 cells or derivatives thereof. Cells have been grown in liquid SC medium of the following composition: 3.25 g yeast nitrogen base (YNB) without amino acids (Difco 291920); 1 g Synthetic Complete Mixture (Kaiser) Drop-Out: Complete (Formedium DSCK1000); 50 ml 20% glucose; 450 mL demineralized water. Phosphate starvation experiments were performed in SC-$P_i$ medium: 3.25 g YNB without amino acids and phosphate, supplemented with KCl (Formedium CYN6703); 1 g Synthetic Complete Mixture (Kaiser) Drop-Out: Complete (Formedium DSCK1000); 50 mL 20% glucose; 450 mL water. These synthetic media were sterile filtered and tested for absence of free $P_i$ by the malachite green assay. Cultures were shaken in capped Erlenmeyer flasks at 150 rpm at 30°C unless stated otherwise.

## Yeast strains and their genetic manipulation

The *S. cerevisiae* strains used in this study are listed and described in Supplementary Information. DNA transformations were carried out with overnight cultures at 30°C in YPD medium (yeast extract-peptone-dextrose) using the following protocol: Cultures (not exceeding $10^7$ cells/mL) were centrifuged at 1800 × *g* for 2 min. 25 µL of pellet was resuspended in 25 µL of LiAc-TE 0.1 M (lithium acetate 0.1 M, Tris 10 mM, EDTA 1 mM). The cell suspension was supplemented with 240 µL of PEG 50%, 36 µL of LiAC 1 M, 5 µL of boiled ssDNA, and the DNA (200 ng of plasmid and/or 0.5–2 mg of PCR product). After 30 min at 30°C, a heat shock was carried out at 42°C for 20 min. Cells were then centrifuged at 1800 × *g* for 2 min, resuspended in 150 µL of water, and grown on selective plates for 2–4 days. Fluorescent protein tags have been obtained using a standard method (*Sheff and Thorn, 2004*), except for Hta2 tagging, which was performed by CRISPR-Cas9. All introduced single mutations, knockouts, and fluorescent protein tags were confirmed by PCR and sequencing.

## Plasmid constructions

The plasmids used to carry out CRISPR-Cas9 genome editing were obtained by cloning hybridized oligonucleotides coding for the sgRNA between the XbaI and AatII restriction sites of the parent plasmid (pSP473, pSP475, or pSP476). The plasmids used to overexpress *VIP1* (pVC97 and pVC114) were obtained by cloning the *VIP1* open reading frame between the BamHI and SalI restriction sites of the parent plasmid (respectively pRS415-GPD and pRS413-GPD) (*Mumberg et al., 1995*). The plasmid used to monitor *prPHO5-GFP* expression (pVC115) was obtained by replacing the *pgk1* promoter with the –1 to –1500 region of the *PHO5* promoter and by replacing the gene for G418 resistance to the coding region of the *LEU2* gene within the pCEV-G1-Km backbone (Addgene).

## Fluorescence microscopy

Cells were grown logarithmically in SC medium for 12–15 hr until they reached a density of $10^7$ cells/mL. For $P_i$ starvation experiments, overnight-grown cells were washed twice, resuspended in SC-$P_i$ medium at a density of ca. $5 \times 10^6$ cells/mL, and incubated at 30°C. Fluorescence images were recorded on a Nikon Eclipse Ti2/Yokogawa CSU-X1 Spinning Disk microscope equipped with two Prime BSI sCMOS cameras (Teledyne Photometrics), a LightHUB Ultra Laser Light (Omicron Laserage), and an Apo TIRF 100×/1.49 Oil lens (Nikon).

Time-lapse experiments were performed on a Nikon Eclipse Ti2 inverted microscope equipped with a Prime BSI sCMOS camera (Teledyne Photometrics), a Spectra X Light Engine (Lumencor), a Plan Apo $\lambda$ 100×/1.45 Oil lens (Nikon). Microscopy chambers were made using sticky-Slides VI $^{0.4}$ (Ibidi) glued onto concanavalin A-coated coverslips (Knittel glass, 24×60×0.13–0.17 mm). The cell suspension (200 µL, $10^7$ cells/mL) was added to the chamber. After 30 s of sedimentation, a flow of SC medium was applied using the microfluidic flow controller AF1 Mk2 (Elveflow) coupled to a MUX distributor (Elveflow). After 2 min, the flow was switched to SC-$P_i$ medium, and imaging was started. After further 2 min, the flow was stopped, and the time lapse was continued by recording one image every 5 min. The temperature was kept at 30°C using the stage-top incubator Uno-Controller (Okolab).

## Western blots

Five mL of cells grown overnight in SC medium (until ca. $10^7$ cells/mL) was centrifuged (1800 × $g$, 2 min). Cells were resuspended in 1 mL of lithium acetate (2 M) and centrifuged as before. Cells were resuspended in 1 mL of sodium hydroxide (0.4 M) and kept on ice for 5 min. After centrifugation (1800 × $g$, 2 min), cells were resuspended in 100 µL of SDS-PAGE sample buffer and boiled for 5 min. After centrifugation under the same conditions, the supernatant was collected, and the protein content was measured using a NanoDrop 1000 Spectrophotometer (Witec). Supernatant concentrations were adjusted, and the samples were loaded on SDS-polyacrylamide gels.

## Inositol pyrophosphate quantification

Cells were logarithmically grown in SC medium overnight to reach a density of $10^7$ cells/mL. One mL of cell culture was mixed with 100 µL of 11 M perchloric acid (Roth HN51.3). After snap-freezing in liquid nitrogen, the samples were thawed and cell debris was removed by centrifugation (15,000 × $g$, 3 min, 4°C). Titanium dioxide beads (Titansphere TiO₂ beads 5 mm, GL Sciences 5020-75000) were pretreated by washing them once with water, once with 1 M perchloric acid, and finally resuspending them in 200 µL 1 M perchloric acid. The cell supernatant was mixed with 1.5 mg of beads and incubated with shaking or on a rotating wheel for 15 min at 4°C. The beads were collected by centrifugation (15,000 × $g$, 1 min, 4°C) and washed twice with 1 M perchloric acid. Inositol phosphates were eluted by 300 µL of 3% ammonium hydroxide and the eluate was evaporated using a speed-vac concentrator at 42°C and 2000 rpm overnight. The pellet was resuspended using 20 µL of water. CE-ESI-MS analysis was performed on an Agilent 7100 CE coupled to triple quadrupole mass spectrometer (QqQ MS) Agilent 6495c, equipped with an Agilent Jet Stream ESI source. Stable CE-ESI-MS coupling was enabled by a commercial sheath liquid coaxial interface, with an isocratic liquid chromatography pump constantly delivering the sheath liquid.

All experiments were performed with a bare fused silica capillary with a length of 100 cm and 50 µm internal diameter. 35 mM ammonium acetate titrated by ammonia solution to pH 9.7 was used as background electrolyte. The sheath liquid was composed of a water-isopropanol (1:1) mixture, with a flow rate of 10 µL/min. 15 µL inositol pyrophosphate extracts was spiked with 0.75 µL isotopic internal standards mixture (2 µM [$^{13}C_6$]1,5-IP$_8$, 10 µM [$^{13}C_6$]5-IP$_7$, 10 µM [$^{13}C_6$]1-IP$_7$, and 40 µM [$^{13}C_6$] IP$_6$). Samples were introduced by applying 100 mbar pressure for 10 s (20 nL). A separation voltage of +30 kV was applied over the capillary, generating a constant CE current at around 19 µA.

The MS source parameters setting were as follows: nebulizer pressure was 8 psi, sheath gas temperature was 175°C, and with a flow at 8 L/min, gas temperature was 150°C and, with a flow of 11 L/min, the capillary voltage was –2000 V with nozzle voltage 2000 V. Negative high-pressure RF and low-pressure RF (ion funnel parameters) was 70 V and 40 V, respectively. Parameters for MRM transitions are in *Table 1*.

## Spectrofluorimetry

Cells were grown overnight to a density of $5×10^6$ cells/mL in SC medium. They were then resuspended in SC-$P_i$ medium after two washing steps. For each timepoint, the concentration of the cell suspension was measured and adjusted to $10^7$ cells/mL before fluorescence measurement. The fluorescence of yEGFP ($\lambda_{ex}$: 400 nm; $\lambda_{em}$: 500–600 nm; $\lambda_{cutoff}$: 495 nm) and mCherry ($\lambda_{ex}$: 580 nm; $\lambda_{em}$: 600–700 nm; $\lambda_{cutoff}$: 590 nm) were measured on a Molecular Devices SpectraMax Gemini EM microplate fluorimeter at 30°C.

## Analysis of inositol pyrophosphate in *C. neoformans* and *S. pombe*

*C. neoformans grubii* wildtype cells were grown in liquid SC medium to logarithmic phase. They were sedimented (3000 × *g*, 1 min, 4°C), washed twice with phosphate-free medium, aliquoted, and used to inoculate four 25 mL cultures in SD medium lacking phosphate. The inoculum was adjusted (starting $OD_{600}$ at 0.9, 0.6, 0.4, 0.3, etc., respectively) such that, after further incubation for 0–240 min, all samples had similar $OD_{600}$ at the time of harvesting. The sample for the 0 min timepoint was taken from the culture in SC medium. The starvation time was counted beginning with the first wash. At the indicated timepoints, the $OD_{600}$ of the culture was measured. A 4 mL sample was collected, supplemented with 400 µL of 11 M perchloric acid, frozen in liquid nitrogen, and kept at –80°C. The samples were thawed, and inositol pyrophosphates were extracted using 6 mg $TiO_2$ beads due to the higher number of cells. For *S. pombe*, wildtype cells were prepared in a similar way to *C. neoformans*. Three mL of culture was collected, supplemented with 300 µL of 11 M perchloric acid, and frozen in liquid nitrogen. All further steps were as described above for *S. cerevisiae* samples.

## Acknowledgements

We thank Julianne Djordjevic, Sophie Martin, and John York for strains and discussion, Dorothea Fiedler for [13]C-labeled inositol pyrophosphate standards, and Andrea Schmidt for technical assistance. This study was supported by grants from SNSF (CRSII5_170925) and ERC (788442) to AM, from HFSP (LT000588/2019) to G-DK, and from the German Research Foundation (DFG) under Germany's Excellence Strategy (CIBSS - EXC-2189 - Project ID 390939984) to HJJ.

## Additional information

### Funding

| Funder | Grant reference number | Author |
| --- | --- | --- |
| Swiss National Science Foundation | CRSII5_170925 | Andreas Mayer |
| ERC | 788442 | Andreas Mayer |
| HFSP | LT000588/2019 | Geun-Don Kim |
| DFG | 390939984 | Henning J Jessen |

The funders had no role in study design, data collection and interpretation, or the decision to submit the work for publication.

### Author contributions

Valentin Chabert, Software, Investigation, Methodology, Writing – original draft, Writing – review and editing; Geun-Don Kim, Investigation, Methodology, Writing – review and editing; Danye Qiu, Guizhen Liu, Lydie Michaillat Mayer, Muhammed Jamsheer K, Investigation, Methodology, Writing – original draft, Writing – review and editing; Henning J Jessen, Formal analysis, Funding acquisition, Investigation, Methodology, Writing – original draft, Project administration, Writing – review and editing; Andreas Mayer, Conceptualization, Formal analysis, Supervision, Methodology, Writing – original draft, Project administration, Writing – review and editing

### Author ORCIDs

Valentin Chabert http://orcid.org/0000-0002-0797-2481
Geun-Don Kim http://orcid.org/0000-0002-3580-2471
Guizhen Liu http://orcid.org/0000-0003-1748-6687
Muhammed Jamsheer K http://orcid.org/0000-0002-2135-8760
Andreas Mayer https://orcid.org/0000-0001-6131-313X

Reviewer #1 (Public Review): https://doi.org/10.7554/eLife.87956.3.sa1
Reviewer #2 (Public Review): https://doi.org/10.7554/eLife.87956.3.sa2
Reviewer #3 (Public Review): https://doi.org/10.7554/eLife.87956.3.sa3

Author Response https://doi.org/10.7554/eLife.87956.3.sa4

## Additional files

### Supplementary files

• MDAR checklist

• Supplementary file 1. This supplementary file contains information on the genetic constitution of the strains used in this study, their origin, and on the plasmids and primers used to generate them. (a) List of strains. (b) List of plasmids. (c) List of primers used for genetic manipulations.

### Data availability

The structure predictions shown in the manuscript and the script used for image analysis have been deposited at Figshare under the https://doi.org/10.6084/m9.figshare.c.6700281.

The following dataset was generated:

| Author(s) | Year | Dataset title | Dataset URL | Database and Identifier |
|---|---|---|---|---|
| Chabert V, Mayer A | 2023 | Methods for automated image analysis used by Chabert et al. (elife 2023) | https://doi.org/10.6084/m9.figshare.c.6700281 | figshare, 10.6084/m9.figshare.c.6700281 |

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
