## [Editor Report · eLife assessment]

This **fundamental** study describes the mechanisms for regulation of the phosphate starvation response in baker's yeast, clarifies the interpretations of prior data, and suggests a unifying mechanism across eukaryotes. The study provides **compelling** data, based on biochemical analyses, protein localization by fluorescence, and genetic approaches that 1,5-InsP8 is the phosphate nutrient messenger in yeast.

---

## [Referee Report · Reviewer #1 (Public Review)]

Recent studies in plants and human cell lines argued for a central role of 1,5-InsP8 as the central nutrient messenger in eukaryotic cells, but previous studies concluded that this function is performed by 1-InsP7 in baker's yeast. Chabert et al now performed an elegant set of capillary electrophoresis coupled to mass spectrometry time course experiments to define the cellular concentrations of different inositol pyrophosphosphates (PP-InsPs) in wild-type yeast cells under normal and phosphate (Pi) starvation growth conditions. These experiments, in my opinion, form the center of the present study and clearly highlight that the levels of all major PP-InsPs drop under Pi starvation, with the 1,5-InsP8 isomer showing the most rapid changes.

The analysis of known mutants in the PP-InsP biosynthetic pathways furthermore demonstrate that loss-of-function of the PPIP5K enzymes Kcs1 and Vip1 result in a loss of 1,5-InsP8 and a hyperaccumulation of 5-InsP7, respectively. In line with this, loss-of-function of known PP-InsP phosphatases Ddp1 and Swi14 result in hyperaccumulation of either 1- or 5-InsP7, as anticipated from their in vitro substrate specificities. These experiments are of high technical quality and add to our understanding of the kinetics of PP-InsP metabolism/catabolism in yeast.

Next, the authors use changes in subcellular localisation of the central transcription factor Pho4 to assay at which time point after onset of Pi starvation the PHO pathway becomes activated. The early onset of the response, the behavior of the kcs1D mutant and of the ksc1D/vip1D all strongly argue for 1,5-InsP8 as the central nutrient messenger. I find this part of the manuscript well argued, nicely correlating PP-InsP levels, dynamics and the different mutant phenotypes.

The third part of the manuscript is a structure-function study of the CDK inhibitor Pho81, basically using a reverse genetics approach. This analysis demonstrates at the genetic level that the Pho81 SPX domain is required for activation of the PHO pathway. Next, the authors design point mutations that should block either interaction of Pho81-SPX with 1,5-InsP8 or interaction of Pho81 with the Pho80/Pho85 complex. In my opinion, these data can only provide limited insight into the molecular mechanism, as no complementary in vitro binding assays / in vivo co-IP experiments with the wild-type and mutant forms of Pho81 are presented. This seems to be due to technical limitations in recombinantly expressing and purifying the respective Pho81 protein for in vitro PP-InsP binding and protein - protein interaction assays.

Taken together, the work by Chabert et al, reinvestigates and clarifies the activation of the yeast PHO pathway by PP-InsP nutrient messengers and their cellular SPX receptors. From this work, a more unified eukaryotic mechanism emerges, in which 1,5-InsP8 represents the central signaling molecule in different species, with conserved SPX receptors sensing this signaling molecule.

---

## [Referee Report · Reviewer #2 (Public Review)]

The manuscript by Chambert et al. describes a thorough and careful characterization of inositol pyrophosphate isomers and the PHO pathways in different genetic backgrounds in *S. cerevisiae*. The paper ultimately arrives at a proposed model in which the inositol pyrophosphate 1,5-IP8 signals phosphate abundance to SPX-domain containing proteins. To arrive at their conclusion, the authors rely heavily on CE-MS analysis of inositol pyrophosphates in different yeast strains, and monitoring inositol pyrophosphate depletion over time in response to phosphate starvation. This analysis is complemented by different reporter systems of PHO pathway activation, such as Pho4 translocation and Pho81 expression.

The experiments are well-designed and the results interpreted with care. With their findings, the authors demonstrate convincingly, that a previous study by O'Shea and co-workers (reference 15 and 16) had been misleading. Lee et al. claimed that the PHO pathway in *S. cerevisiae* is triggered by an increase in 1-IP7. This claim has been debated heavily in the community, and several groups were not able to reproduce this putative increase of inositol pyrophosphates (references 6, 11, 18). The confusion regarding these discrepancies has been resolved by the current study and is of significant importance to the community.

---

## [Referee Report · Reviewer #3 (Public Review)]

Summary. This study sought to clarify the connection between inositol pyrophosphates (IPPs) and their regulation of phosphate homeostasis in the yeast *Saccharomyces cerevisiae* to answer the question of whether any of the IPPs (1-IP7, 5-IP7, and IP8) or only particular IPPs are involved in regulation. IPPs bind to SPX domains in proteins to affect their activity, and there are several key proteins in the PHO pathway that have an SPX domain, including Pho81. The authors use the latest methodology, capillary electrophoresis and mass spectrometry (CE-MS), to examine the cytosolic concentrations of PP-IPs in wild-type and strains carrying mutations in the enzymes that metabolize these compounds in rich medium and during a phosphate starvation time-course for the wild-type.

Major strengths and weaknesses. The authors have strong premises for performing these experiments: clarifying the regulatory molecule(s) in yeast and providing a unifying mechanism across eukaryotes. They use the latest methodologies and a variety of approaches including genetics, biochemistry, cell biology and protein structure to examine phosphate regulation. Their experiments are rigorous and well controlled, and the story is clearly told. The consideration of physiological levels of IPPs throughout the study was critical to interpretation of the data and a strength of the manuscript. The investigation of the structure of Pho81, its regulation by IPPs, and its interactions with Pho80 provide a vivid model for regulation.

Appraisal. The authors achieved their goal of determining the mechanistic details for phosphate regulation, revising the prior model with new insights. Additionally, they provided strong support for the idea that IP8 regulates phosphate metabolism across eukaryotes - including animals and plants in addition to fungi.

Impact. This study is likely to have broad impact because it addresses prior findings that are inconsistent with current understanding, and they provide good reasoning as to how older methods were inadequate.

---

## [Author Response]

The following is the authors’ response to the original reviews.

**Reviewer #1 (Recommendations For The Authors):**
Introduction: "In plants, IP7 and IP8 decrease upon Pi starvation and mutants lacking either of the enzymes necessary for their synthesis induce the phosphate starvation response, leaving it open whether IP7 , IP8 , or both, control the Pi starvation program (10, 13, 14)" In plants, ITPK enzymes catalyze the formation of 5-InsP7 from InP6 https://pubmed.ncbi.nlm.nih.gov/34274522/ and VIH enzymes the formation of 1,5-InsP8 from 5-InsP7 https://pubmed.ncbi.nlm.nih.gov/34274522/ and of VIH1/2 https://pubmed.ncbi.nlm.nih.gov/31419530/ results in constitutive Pi starvation responses. vih1/vih2 mutants lack 1,5-InsP8 and hyperaccumulate 5-InsP7 https://pubmed.ncbi.nlm.nih.gov/31436531/ and act directly upstream of the transcription factors PHR1/PHL1 https://pubmed.ncbi.nlm.nih.gov/33452263/
https://pubmed.ncbi.nlm.nih.gov/34857773/, the case for 1,5-InsP8 is settled in plants. I would suggest revising this statement and citing all relevant literature.

We don't see the case for 1,5-IP8 as settled in plants, and none of the papers mentioned above draws this strong conclusion. This may be due to several limitations in the available data. The mentioned studies do not allow to differentiate the effects of 1-IP7 and 1,5-IP8 and, where binding or competition experiments have been performed, e.g. on the transcription factors, the differences in the Kd values for IP7 and IP8 were minor. Furthermore,1,5-IP8 levels and Pi starvation response do not always correlate. IPTK1 mutants, for example, show Pi overaccumulation, and low 5-IP7, but normal 1,5-IP8 (Riemer et al., 2021). Finally, plants are complex organisms with multiple tissue types that serve for accumulating, exporting, transporting or finally consuming Pi. Therefore, correlating inositol pyrophosphate levels from whole-plant extracts with a Pi starvation response is problematic, except if these data could both be obtained from the same cell types or at least tissues.

The comment of the reviewer made us recognize that the complex situation in plants deserves a more detailed coverage and we have therefore adjusted the introduction accordingly.

Results: "We determined the corresponding lysines in Pho81 (Fig. S3), created a point mutation in the genomic PHO81 locus that substitutes one of them, K154, by alanine, and investigated the impact on the PHO pathway."In my opinion, it would be important to test here in a quantitative in vitro binding assay if (i) the SPX domain of Pho81 can bind PP-InsPs including 1,5-InsP8, (ii) if the dissociation constant is in agreement with the cellular levels of 1,5-InsP8 in yeast (compare Fig. 2) and (iii) if the K154A mutation blocks or reduces the binding of 1,5-InsP8. Without such experimentation, I find the statement "this result underlines the efficiency of the K154A substitution in preventing PP-IP binding to the Pho81 SPX domain." to be overly speculative, as no binding experiment has been conducted.

We agree with the comment of the reviewer concerning the overstatement in the phrase. It has been deleted.

As mentioned already in our previous work (Wild et al., 2016), Pho81SPX counts among the SPX domains that we could not express recombinantly. Likewise, full-length Pho81, which would be the relevant object for correlating in vitro binding studies with the cellular concentrations, has not been accessible. Expression in yeast did not provide sufficient material for ITC or other quantitative techniques. Therefore, we refrained from pursuing binding studies. Nevertheless, given the high conservation of the positively charged patch on SPX domains and the fact that, in every case where it has been tested so far, SPX domains showed inositol polyphosphate binding activity, we find it a conservative assumption that the Pho81SPX binds them as well. This is supported by the effects of the binding site mutant, which mimics the effect of ablating IP8 synthesis.

Results: "Inositol pyrophosphate binding to the SPX domain labilizes the Pho81-Pho80 interaction." Again, in the absence of any protein - protein interaction assay I find this statement not to be supported by the experiments outlined in the manuscript. The best way to address this point would be to perform either co-IP or in vitro pull-down experiments between Pho81-SPX and Pho81-85, in the pre- and absence of 1,5-InsP8 and/or using the Pho81 point-mutants described in the text.

Since Pho81 could not be produced recombinantly, neither by us nor by others who worked on this protein previously, quantitative in vitro binding assays are not accessible for now. A simple IP suffers from the problem that Pho81 interacts with Pho85-Pho80 not only through the SPX domain but also through the minimum domain. The latter interaction may be constitutive. Since the main point of the manuscript is not to dissect the exact mechanisms of Pho85-Pho80 regulations, but only to address the point why the postulated inactivation of this kinase by an 1-IP7/minimum domain complex makes no sense, we prefer not to show a profound (and more complex) analysis of how the different Pho81 domains contribute to binding.

To test the potential of the SPX domain for binding Pho85/Pho80 in vivo, we have created a GFP-fusion of the SPX domain of Pho81. This fusion protein localizes mainly to the cytosol when cells are on high-Pi. Upon Pi starvation, it concentrates in the nucleus. This concentration is not observed in pho80 mutant background (New Fig. S7).

In line with this, I would suggest to move the molecular modelling/docking studies from the discussion into the results section and to use these models to design some interface mutations that could be tested in coIP and/or pull-down assays. Alternatively, the authors may choose to omit the discussion section starting with: "Even though the minimum domain is unlikely to function as a receptor for PP-IPs this does not ... and ending with . In sum, multiple lines of evidence support the view that the SPX domain exerts dominant, 1,5-IP8 mediated control over Pho81 activity in response to Pi availability."

We have now moved the modelling data to the Results section.The structure prediction of the interface is experimentally validated. Data on the effect of interface substitutions are already published, although these substitutions had not been recognized as affecting a common interface at the time. Substituting the interface residues either on the side of Pho80 or of Pho81 constitutively activates Pho85-Pho80 kinase and destabilizes its interaction with Pho81. This was shown by Co-IP experiments from cell extracts by Huang et al. We mention the respective substitutions in the manuscript and cite the paper in which their effect on PHO pathway activation had been described.

**Reviewer #2 (Recommendations For The Authors):**
Some points need additional attention by the authors:In general, it would be helpful to introduce abbreviations more thoroughly (certain enzyme names, PA, MD, ...)

We paid more attention to this.

Also in general, the authors may want to think about the nomenclature of inositol pyrophosphates. Given the expansion of PP-IPs that are being detected in different organisms these days it may be a good time to convert to a more precise nomenclature, i.e. 5PP-IP5 instead of 5-IP7; and 1,5(PP)2-IP4, instead of 1,5-IP8. The latter could just be stated once, and then be abbreviated as IP8.

To our understanding the field has not yet come up with a unified nomenclature. Therefore, we prefer to stick with the more practical nomenclature that we have chosen, which also corresponds to what is commonly used in presentations and discussions among colleagues. We have now introduced a sentence making the link to the nomenclature that the reviewer has proposed.

p. 1, Abstract: "negative bioenergetic impacts" - the phrasing seems really vague

Agreed, but we find it difficult to be more explicit and precise in the abstract while remaining concise and not distracting from the main message. This aspect is better explained in the introduction.

p. 3, Significance statement: "... unified model across all eukaryotic kingdoms" While the intended meaning of this wording is better explained in the text later, the phrasing here suggests a more all-encompassing study at hand, instead of a conclusion that fits more closely with established reports from other organisms. Please rephrase.

We have adapted the phrase to avoid this impression.

p. 4: "IPTKs" - are the ITPKs meant here?

Yes, that was a typo.

p. 7, the introduction ends abruptly and could use a concluding sentence.

Done

p.7, "enzymes diphosphorylation either the..."; I understand what the authors are trying to say with diphosphorylating, but the enzymes are phosphorylating a phosphorylated substrate.

Yes. We changed the phrase to "....adding phosphate groups at the 1- or 5-positions....".

p. 7, subtitle "...concentrations and kinetics of..."; kinetics of what? Synthesis/turnover?

We corrected this subtitle

p. 8, with regards to the recovery experiment: Was this recovery determined elsewhere (please cite)? Otherwise it would be beneficial to include an extra figure to illustrate these recoveries in the supplementary information. And do the authors suspect some hydrolysis of IP8 given the lower recovery?

We have now added the experiment testing recovery of IPPs as the new Fig. S1.

p. 9: It is appreciated that the authors point out the concentration of IP6 in *S. cerevisiae*. I found that concentration rather low, and the authors could highlight this a bit more, given their ability to carry our absolute quantification.

This was a leftover from a previous version of the paper. Since the paper does not treat IP6 or lower inositol polyphosphates, we have deleted this phrase.

p. 9, Fig 2: The exponential decay of 5-IP7 is very nicely shown in Figure 2c. But one of the most important discussion points is IP8 being the key controller of the PHO pathway - it would therefore be beneficial for the argument to also show the same kind of graph for IP8 and if possible, fit a function to the data points to better quantify and compare the decay processes (e.g. via "half-life time" of PP-IPs during starvation, in addition to the suggested "critical concentration" which was only discussed for 5-IP7 thus far).

Kinetic resolution is an issue here. The approach shown in Figs. 2 and 5 is not apt to determine a critical concentration of IP8 because the decline upon transfer to starvation conditions is too fast and difficult to relate to the equally rapid induction of the PHO pathway. We shall address this point in a more appropriate setup in a future study.

p.9, Fig 2a: Where does the 5-IP7 come from in the kcs1Δ strain? In the text the authors state that 5-IP7 in kcs1Δ was not detected, but the figure suggests otherwise. Please explain.

Currently, we do not know where these residual signals stem from. One possibility is that they represent other isomers that exist in minor concentrations and that are not resolved from 5-IP7 in CE. We added a sentence to the figure legend to indicate this.

p. 10: "IP8 was undetectable in kcs1Δ and decreased by 75% in vip1Δ. kcs1Δ mutants also showed a 2 to 3-fold decrease in 1-IP7, suggesting that the synthesisof 1-IP7 depends on 5-IP7. This might be explained by assuming that a significant source of 1-IP7 is synthesis of 1,5-IP8 through successive action of Kcs1 and Vip1, followed by dephosphorylation to 1-IP7." - Please specify this statement. Do the authors mean that 1,5-IP8 is only produced transiently below the detection capabilities of the method but that there still is a (reduced) flux from 5-IP7 to 1,5-IP8 to 1-IP7? Otherwise it would seem paradoxical to have a dependency on a non-existing metabolite in that cell line.

This was not clearly expressed. The revised version now says: " ... a 2 to 3-fold decrease in 1-IP7, suggesting that the synthesis of 1-IP7 depends on 5-IP7. This might be explained by assuming that, in the wildtype, most 1-IP7 stems from the conversion of 5-IP7 to 1,5-IP8, followed by dephosphorylation of 1,5-IP8 to 1-IP7.". We hope that this clarifies the matter.

p. 10: "pulse-labeling approaches are not available for PP-IPs." While this statement is correct, a recent paper co-authored by Qui and Jessen showed nice pulse-labeling data for the lower Ips and could be cited here (PMID: 36589890)

Yes, indeed, we should have been more precise here. What we wanted to express was that rapid pulse-labeling methods for following phosphate group turnover were lacking, with a temporal resolution of minutes rather than hours. Existing pulse labeling approaches, including the study mentioned by the reviewer, do not provide that. We have changed the phrase accordingly.

p. 10: continuation of caption of Fig 2: "were extracted [and] analyzed"

Corrected. Thank you.

p. 12: How is 1-IP7 made in the vip1 kcs1 double mutant?

As explained above, we suspect that these may be side products of IPMKs, which accumulate in the absence of vip1 phosphatase.

p. 13, caption to Figure 3: "XXX cells were analyzed" please replace the place holder XXX.

Done. Thank you.

p. 13, Fig 3B, C, D and p. 50, Fig. S4: On screen the contrast between the different shades of grey of the bars are just visible enough, but not on paper, I suggest using a higher contrast/ different colouring scheme.

We enhanced the contrast.

p. 24, 25, Fig 7.: I could not really appreciate the AlphaFold part, and found it unnecessary. No docking or molecular dynamics simulations were carried out here, and it was not clear to me what information should be gleaned from this part.

Following this comment, we have modified the respective part of the text. This part refers to a publication from the O'Shea lab (Nat. Chem Biol. 4,25) proposing the model that 1-IP7 and the Pho81 minimum domain bind competitively to the active site of Pho85 to inhibit its kinase activity. Modeling of complexes between Pho81, Pho80 and Pho85, which we present in the manuscript, rather suggests binding of the minimum domain to a groove in Pho80. This is important because it provides a viable alternative model for the action of the minimum domain. It suggests the minimum domain as a constitutive linker that attaches Pho80 to Pho85. Importantly, this model accounts perfectly for the results of previous random mutagenesis studies on Pho80 and on the minimum domain, which had independently identified both the Pho80 groove and the minimum domain residues that bind it in the prediction as critical residues for inhibition of Pho85, and for integrity of the Pho85/Pho80/Pho81 complex. We find this alternative explanation for Pho85-Pho80 regulation by Pho81, which we can derive by combining the predictions with already published experimental data, an important element to re-evaluate the relevance of 1-IP7 in PHO pathway regulation and resolve one of the existing discrepancies.

p. 28: No experiments were carried out with plants or mammals. The relevance for plants or mammalian systems therefore seems to be overstated at this point in time.

We are not quite sure how to interpret this remark. We do not claim that our data support a role for IP8 in mammals and plants. But we refer to and cite studies providing the strongest evidence in favor of it in these systems. The relevance of our current study relies in refuting seemingly strong evidence from yeast, which had been diametrically opposed to the data obtained in plants and mammals. The revision of the situation in yeast now paves the way to drawing a coherent concept for fungi, plants and mammals. We feel that this is important and should be underlined.

p. 31: "300 mL of 3% ammonium" - 300 µL?

Yes. Thank you.

p. 45, CE-ESI-MS parameters: "1IP8"

Corrected.

p. 47: Figure S1: Please include more experimental details in the caption and/or methods section. Was a similar analysis software used as e.g. Figure S2 (NIS Elements Software)? Please also include all the analysis software in the Methods section under "fluorescence microscopy". Unless these additional experimental details already clarify the following point: Can the authors briefly comment on why the morphological determination in S1 requires trypan blue staining while in later experiments the yeast cells are readily recognized by the software in "simple" brightfield images?

Trypan blue staining is not strictly required for this. It is just a simple method to fluorescently stain the cell wall. There are many other ways of delineating the cells. It could also have been done in a brightfield image.

We updated the figure legend to better describe how these measurements were done and deposited the script and training file on figshare.

p. 48: "can be downloaded from ******" please insert the link once the script is available online.

It has been deposited at Figshare under DOI 10.6084/m9.figshare.c.6700281

**Reviewer #3 (Recommendations For The Authors):**
1. Italicize the scientific names of the organisms; this was inconsistent throughout the manuscript. Also, gene names should be italicized; this was also inconsistent (e.g., p.12 "... did not induce the PHO84 and PHO5 [sic] promoters...).

Done

2. Summary of the Figure 2A data in the text (p.9) probably has swapped the determined concentrations for 1-IP7 and IP8 (0.3 µM or 0.5 µM) as compared with the data figure.

Yes, indeed. We have corrected this.

3. Figure 2A: which of the mutant PP-IP levels are significantly different from the WT control?

We have now added asterisks to indicate the significance for every mutant.

4. In the discussion on the data (Fig. 2A), I was tripped up by the verb tense in this phrase "5-IP7 has not been detected in the kcs1Δ mutant and 1-IP7 has been strongly reduced..."; I think you want to use the past tense "was" in both cases [as is used in the next sentence]. It made me wonder if there was a difference in the detection of 5-IP7 and IP8 in the kcs1Δ mutant, you could detect 5-IP7 but not IP8; if so, where did the 5-IP7 come from?

We have corrected the tense. Thank you for highlighting this. For the residual inositol pyrophosphate signal in kcs1Δ. We do not know its origin. One possibility, which we now mention in the text, is that it stems from IPMK side activity. It should be underlined that all signals disappear upon PI starvation.

Figure 2C, include the data points that the lines are built from (suggestion).

We refrained from that for the line graphs. For reasons of consistency, we should do this for every line graph. If we did that, Fig. 4B would become quite hard to read.

6. Figure 3B-D, please check that the stipples or hatches are in the figure - the printed copy lacked them although I could see them in the electronic version; this was also true for Figures 5 and 6 (I do not know if it is a printer issue, but other hatches were visible: e.g., not seen in S4 but seen in S5).

They are visible in our copies, also after printing. They may have been lost during file conversion at the journal.

7. The text description of the Pho4-yEGFP, Pho5-yEGFP and Pho84-yEGFP says that the kcs1Δ mutant "showed Pho4-yEGFP constitutively in the nucleus already ... and PHO5 and PHO84 were activated". However, the data is more complex than that: whereas the localization of Pho4-yEGFP is constitutively nuclear, there is a higher basal (repressed) expression of both Pho5 and Pho84 as well as increased expression of both proteins under -Pi conditions. What accounts for the increased expression when Pho4 is already nuclear? This is also seen in the vip1Δ kcs1Δ mutant.

We agree with the reviewer, but we cannot explain this effect with certainty. One possibility could be a wider dysregulation of Pi metabolism in kcs1 mutants. To name a few possibilities: Wildtype cells have polyphosphate reserves that are gradually mobilized during the first hours of P-starvation. kcs1 mutants don't have those and might fall into a "deeper" state of starvation faster. It should be kept in mind that the starvation response is also regulated at the level of chromatin structure, and by antisense transcripts. The influence of kcs1 on these processes is unclear.

8. Figure 9 legend: please add a definition of the MP region (in red) and include it more explicitly in the described model.

We now mention the relevant region also in the legend and have labeled the relevant regions in the images (Huang et al., 2001).

9. Figure S2 legend: information is missing (downloading link).

It has been deposited at Figshare under DOI 10.6084/m9.figshare.c.6700281

10. Figure S4 and S5, missing statistics.

They have been added to the new Fig. S6, which interprets differences between strains and conditions. Fig. S4 (now S3) shows timecourses of IPPs down to zero. Adding statistics for all pairwise differences between the timepoints would be almost an overkill.